

# Carbon emission and export from Ket River, western Siberia

**Artem G. Lim[1], Ivan V. Krickov[1], Sergey N. Vorobyev[1],**

**Mikhail A. Korets[2], Sergey Kopysov[1],**

**Liudmila S. Shirokova[3,4], Jan Karlsson[5], and Oleg S. Pokrovsky[3*]**

[1]*BIO-GEO-CLIM Laboratory, Tomsk State University, Tomsk, Russia*
[2] *V.N. Sukachev Institute of Forest of the Siberian Branch of Russian Academy of Sciences – separated department of the KSC SB RAS, Krasnoyarsk, 660036, Russia*
[3] *Geosciences and Environment Toulouse, UMR 5563 CNRS, 14 Avenue Edouard Belin 31400 Toulouse, France*
[4] *N. Laverov Federal Center for Integrated Arctic Research, Russian Academy of Sciences, Arkhangelsk, Russia*

[5]*Climate Impacts Research Centre (CIRC), Department of Ecology and Environmental Science, Umeå University, Linnaeus väg 6, 901 87 Umeå, Sweden.*

Key words: $CO_2$, C, emission, boreal, river, export, landscape, Siberia

* email: oleg.pokrovsky@get.omp.eu



**Abstract**
Despite recent progress in the understanding of the carbon (C) cycle of Siberian permafrost-affected
rivers, spatial and seasonal dynamics of C export and emission from medium-size rivers remain poorly
unknown. Here we studied one of the largest tributaries of the Ob River, the Ket River (watershed = 94,000
km²) which drains through virtually pristine dense taiga forest of the boreal zone in western Siberian Lowland
(WSL). We combined continuous in-situ measurements of carbon dioxide ($CO_2$) concentration and flux
($FCO_2$), with methane ($CH_4$), organic and inorganic C (DOC and DIC, respectively), particulate organic C
and total bacterial concentrations over a 834-km transect of the Ket River main stem and its 26 tributaries
during spring flood and 12 tributaries during summer baseflow. The $CO_2$ concentration was lower and less
variable in the main stem (2000 to 2500 µatm) compared to that in tributaries (2000 to 5000 µatm). The
methane concentrations in the main stem and tributaries was a factor of 300 to 1900 (flood period) and 100
to 150 (baseflow period) lower than that of $CO_2$. The $FCO_2$ ranged from 0.4 to 2.4 g C $m^{-2}$ $d^{-1}$ in the main
channel and from 0.5 to 5.0 g C $m^{-2}$ $d^{-1}$ in the tributaries, being the highest during August in tributaries and
weakly dependent on season in the main channel. Only during summer baseflow, the DOM aromaticity,
bacterial number, and needleleaf forest coverage of the watershed positively affected $CO_2$ concentrations and
fluxes. We hypothesize that the relatively low variability in $FCO_2$ is due to flat homogeneous (bog and taiga
forest) landscape that results in long water residence times and stable input of allochthonous DOM, which
dominate the $FCO_2$. In summer baseflow, the DIC input from deeper flow paths might also contribute to $CO_2$
emission. The open water period (May to October) C emission from the Ket River basin was estimated to
127±11 Gg C $y^{-1}$ which is lower than the lateral C export during the same period. Although this estimated C
emissions contain uncertainties, stressing the need of better constrained $FCO_2$ and water coverage across
seasons, we considered it conservative which emphasize the important role of WSL rivers for release of $CO_2$
to the atmosphere.








**Introduction**

Assessment of greenhouse gas (GHG) emission from rivers is crucially important for understanding the C cycle under various climate change scenarios (Campeau and del Giorgio, 2014; Chadburn et al., 2017; Tranvik et al., 2018; Vonk et al., 2019; Vachon et al., 2020). Rivers receive terrestrial C and process and emit a significant share of this C during transit to the sea (Liu et al., 2022). Quantifications of riverine C emissions are sufficiently robust for relatively well studied regions of the world such as the European and N American boreal zone (Dawson et al., 2004; Dinsmore et al., 2013; Wallin et al., 2013; Leith et al., 2015), or Arctic and subarctic rivers of Alaska (Striegl et al., 2012; Crawford et al., 2013; Stackpoole et al., 2017). Despite significant progress in assessing riverine $pCO_2$ in previously under-represented or ignored regions such as lotic systems of Asia (Ran et al., 2015, 2017; Varol and Li, 2017) or South America (Almeida et al., 2017), these studies generally use a combination of pH and alkalinity (DIC) to calculate the $pCO_2$ instead of direct in-situ measurements, alike the studies of global emissions (Raymond et al., 2013; Lauerwald et al., 2015). In this regard, regional high spatial resolution measurements of $CO_2$ concentration and fluxes of under-represented regions are needed.

High latitude regions are important in this respect given their large C stocks, partly located in the permafrost, and the observed and projected warming (Turetsky et al., 2020). This is especially true for Siberia, hosting large C stocks in soils and wetlands intersected by extensive river networks that deliver majority of water and C to the Arctic Ocean (Feng et al., 2013). There has been substantial progress in quantification of carbon (C) transport and emissions from Siberian permafrost-affected rivers (Lobbes et al., 2000; Raymond et al., 2007; Cooper et al., 2008; Semiletov et al., 2011; Feng et al., 2013; Griffin et al., 2018; Wild et al., 2019). However, spatial and seasonal features of C export and emission from tributaries of Siberian rivers are still remain poorly known. Existing data (Denfeld et al., 2013; Serikova et al., 2018; Karlsson et al., 2021; Vorobyev et al., 2021) suggest that C (predominantly as $CO_2$) emissions from Siberian rivers can vary largely over space and time. Such high variations do not allow reliable quantitative assessment of C emission and integrating these values into regional and global C models.

In order to better understand and constrain the magnitude of C emission from Siberian rivers we studied the Ket River (watershed 94,000 km²), a typical tributary of the Ob River in western Siberia. The Ob





river is the largest (in terms of watershed area) Siberian river and drains large pristine territories of taiga forest
and bogs. The catchment of Ob includes extensive regions of permafrost but a major part of it (80 %) is
situated in the permafrost-free zone of which very few data exist on riverine C emissions (Karlsson et al.,
2021). The Ket river drains permafrost-free western Siberian forest and wetlands with almost no human
activity, thus serving a representative system for understanding C cycling of permafrost-free rivers of an
underrepresented region of the world. We followed, via a boat routing over the main stem and main tributaries
of the river, the in-situ $CO_2$ concentrations combined with discrete regular sampling for dissolved $CH_4$, DOC,
DIC, total bacterial number and particulate organic matter. These measurements were complemented with
regular floating chamber measurements of $CO_2$ emission fluxes. We performed these observations during two
main open water seasons of the year - the peak of the spring flood and the end of the summer baseflow. Our
first objective was to quantify the difference in C concentration and emission during two seasons for the main
steam and the tributaries and to relate these differences to main physico-chemical parameters of the water
column and physio-geographical parameters (land cover) of the river watersheds. Our second objective was
to obtain total C emission flux from the river watershed area and compare it to lateral export yield of dissolved
and particulate carbon.

**2. Study Site, Materials and Methods**
*2.1. Ket River and its tributaries*

The Ket River main stem and its 26 tributaries sampled in this study include watersheds of distinct

sizes (catchment area ranged from 94,000 at the Ket's mouth to 20 km² of smallest tributary), but rather
similar lithology, climate and vegetation (**Fig. 1, Table S1**). This poorly accessible river basin is fully pristine
(50 % forest, 40 % wetlands), and has almost no agricultural and forestry activity. The watershed of Ket has
very low population density (0.27 person km$^{-2}$) and lacks road infrastructure due to absence of hydrocarbon
exploration activity. In this regard, this river can serve as a model for medium size bog-forest rivers of the
western Siberia Lowland and results obtained from this watershed can be extrapolated to much larger
territory, comprising about 1 million km$^2$ of permafrost-free taiga forest and bog regions of the southern part
of WSL.





The mean annual air temperatures (MAAT) is  -0.6..-0.9 °C and the mean annual precipitation is 520
mm $y^{-1}$ in the central part of the basin. The lithology of this part of western Siberian lowland is dominated by
Pleistocene silts and sands with carbonate concretions overlayed by quaternary deposits (loesses, fluvial,
glacial and lacustrine deposits). The dominant soils are podzols in forest areas and histosols in peat bog
regions.
The peak of annual discharge in 2019 occurred in the end of May; in August, the discharge was 3 to
5 times smaller (**Fig. 1**). From May 18 to May 28, 2019, and from August 30 to September 2, 2019, we started
the boat trip in the middle course of the Ket River (Beliy Yar), and moved, first, 475 km upstream the Ket
river till its most headwaters, and then moved 834 km downstream till the river mouth, with an average speed
of 20 km $h^{-1}$. We stopped each 30-50 km along the Ket River and sampled for major hydrochemical
parameters, GHG, river suspended matter and total bacterial number of the main stem. We also moved several
km upstream of selected tributaries to record $CO_2$ concentrations for at least 1 h and to sample for river
hydrochemistry. At several occasions during spring flood, we monitored $CO_2$ concentration and performed
chamber measurements in the main stem and tributaries during both day and night time period.

*2.2. $CO_2$ and $CH_4$ concentrations and $CO_2$ fluxes by floating chambers*
Surface water $CO_2$ concentration was measured continuously, *in-situ* by deploying a portable infrared
gas analyzer (IRGA, GMT222 CARBOCAP® probe, Vaisala®; accuracy ± 1.5%) of two ranges (2 000 and
10 000 ppm) as described in previous work of our group on the Lena River (Vorobyev et al., 2021). The probe
was enclosed within a waterproof and gas-permeable membrane. For this, we used a protective expanded
polytetrafluoroethylene (PTFE) tube or sleeve that is highly permeable to $CO_2$ but impermeable to water
(Johnson et al., 2009). During the sampling, the sensor was left to equilibrate in the water for 10 minutes
before measurements were recorded. The sensor was placed into a tube which was submerged 0.5 m below
the water surface. A Campbell logger was connected to the system allowing continuous recording of the $CO_2$
concentration, water temperature and pressure every minute over 10 minute intervals yielding 732 individual
$p$$CO_2$, water temperature and pressure values. The $CO_2$ concentrations in the Ket River tributaries included
between 10 and 20 individual $p$$CO_2$ readings for each tributary (250 measurements in total). In addition to



continuous *in-situ* $CO_2$ measurements, we estimated $pCO_2$ via measured pH and DIC values, using the set of
constants typically applied for riverine $pCO_2$ estimation in organic-rich waters (Cai and Wang, 1998;
DelDuco and Xu, 2017). The U-test (Mann-Whitney) demonstrated a lack of significant difference in $CO_2$
concentrations measured by Vaissala and calculated from the pH and DIC of the river water.
For $CH_4$ analyses, unfiltered water was sampled in 60-mL Serum bottles, closed without air bubbles
using vinyl stoppers and aluminum caps and immediately poisoned by adding 0.2 mL of saturated $HgCl_2$ via
a two-way needle system. Headspace was created in the laboratory and $CH_4$ concentrations were analyzed
using a Bruker GC-456 gas chromatograph (GC) equipped with flame ionization and thermal conductivity
detectors. Further details of $CH_4$ analyses are described elsewhere (Serikova et al., 2019; Vorobyev et al.,

2021).

The $CO_2$ fluxes were measured by using two floating $CO_2$ chambers equipped with non-dispersive
infrared SenseAir® $CO_2$ loggers (Bastviken et al., 2015), at each of the 7 (spring flood) and 6 (summer
baseflow) sampling location of the main stem and 26 tributaries following the procedures described elsewhere
(Serikova et al., 2019; Krickov et al., 2021). In addition to *in-situ* chamber measurements, the $CO_2$ flux was
calculated from measured $CO_2$ concentration using standard approaches (Guérin et al., 2007; Wanninkhof,
1992; Cole and Caraco, 1998). The value of $K_T$ (gas transfer velocity) was calculated in two ways - assuming
zero wind speed and the actually measured wind speed at the site of sampling or at the nearest meteo-station
located in the Belyi Yar town, middle course of the Ket River. For comparison with previous estimates, we
also used a gas transfer velocity of 4.46 m $d^{-1}$ measured in the 4 largest rivers of Western Siberia Lowalnd
(WSL) in June 2015 (Ob', Pur, Pyakupur and Taz rivers, Karlsson et al., 2021) which is representative for
large lowland rivers (Alin et al., 2011; Beaulieu et al., 2012).

*2.3. Chemical analyses of the river water*
The dissolved oxygen (CellOx 325; accuracy of ±5%), specific conductivity (TetraCon 325; ±1.5%),
and water temperature (±0.2 °C) were measured in-situ at 20 cm depth using a WTW 3320 Multimeter. The
pH was measured using portable Hanna instrument via combined Schott glass electrode calibrated with NIST
buffer solutions (4.01, 6.86 and 9.18 at 25°C), with an uncertainty of 0.01 pH units. The temperature of buffer



solutions was within ± 2°C of that of the river water. The water was sampled in pre-cleaned polypropylene
bottle from 20-30 cm depth in the middle of the river and immediately filtered through disposable single-use
sterile Sartorius filter units (0.45 µm pore size). The first 50 mL of filtrate was discarded. The DOC and
Dissolved Inorganic Carbon (DIC) were determined by a Shimadzu TOC-VSCN Analyzer (Kyoto, Japan)
with an uncertainty of 3% and a detection limit of 0.1 mg/L. Blanks of MilliQ water passed through the filters
demonstrated negligible release of DOC from the filter material. The SUVA was measured via ultraviolet
absorbance at 254 nm using a 10-mm quartz cuvette on a Bruker CARY-50 UV-VIS spectrophotometer.
The concentration of C and N in suspended material (Particulate Organic Carbon and Nitrogen (POC
and PON, respectively)) was determined via filtration of 1 to 2 L of freshly collected river water (at the river
bank or in the boat) with pre-weighted GFF filters (47 mm, 0.45 µm) and Nalgene 250-mL polystyrene
filtration units using a Mityvac® manual vacuum pump. Particulate C and N were measured using catalytic
combustion with Cu-O at 900°C with an uncertainty of ≤ 0.5% using Thermo Flash 2000 CN Analyzer at
EcoLab, Toulouse. The samples were analyzed before and after 1:1 HCl treatment to distinguish between
total and inorganic C; however the ratio of $C_{organic}$ : $C_{carbonate}$ in the river suspended matter (RSM) was always
above 20 and the contribution of carbonate C to total C in the RSM was equal in average 0.3±0.3% (2 s.d., n
= 30).
Total microbial cell concentration was measured after sample fixation in glutaraldehyde, by a flow
cytometry (Guava® EasyCyteTM systems, Merck). Cells were stained using 1 µL of a 10 times diluted SYBR
GREEN solution (10000x, Merck), added to 250 µL of each sample before analysis. Particles were identified
as cells based on green fluorescence and forward scatter (Marie et al., 2001).

*2.5. Landscape parameters and water surface area of the Ket River basin*
The physio-geographical characteristics of the 26 Ket tributaries and the 7 points of the Ket main stem
(**Table S1, Fig. S1**) were determined by applying available digital elevation model (DEM GMTED2010),
soil, vegetation and lithological maps. The landscape parameters were typified using TerraNorte Database of
Land Cover of Russia (Bartalev et al., 2020; http://terranorte.iki.rssi.ru). This included various type of forest
(evergreen, deciduous, needleleaf/broadleaf), grassland, tundra, wetlands, water bodies and riparian zones.



The climate parameters the watershed were obtained from CRU grids data (1950-2016) (Harris et al., 2014)
and NCSCD data (Hugelius et al., 2013; doi:10.5879/ecds/00000001), respectively, whereas the biomass and
soil OC content were obtained from BIOMASAR2 (Santoro et al., 2010) and NCSCD databases. The
lithology layer was taken from GIS version of Geological map of the Russian Federation (scale 1 : 5 000 000,
http://www.geolkarta.ru/). We quantified river water surface area using the global SDG database with 30 m²
resolution (Pekel et al., 2016) including both seasonal and permanent water for the open water period of 2019
and for the multiannual average (reference period 2000-2004). We also used a more recent GRWL Mask
Database which incorporates first order wetted streams (Allen and Pavelsky, 2018).

*2.6. Data analysis*
Carbon concentrations and fluxes for all dataset were tested for normality using a Shapiro-Wilk test.
In case of the data were not normally distributed, we used non-parametric statistics. Comparisons of GHG
parameters in the main stem and tributaries during two sampling seasons were conducted using a non-
parametric Mann Whitney test at a significance level of 0.05. For comparison of unpaired data, a non-
parametric H-criterion Kruskal-Wallis test was used to reveal the differences between different study sites.
The Pearson rank order correlation coefficient ($p < 0.05$) was used to determine the relationship between $CO_2$
concentrations and emission fluxes and main landscape parameters of the Ket River tributaries, as well as
other potential drivers such as pH, $O_2$, water temperature, specific conductivity, DOC, DIC, particulate carbon
and nitrogen, and total bacterial number.

**3. Results**
*3.1. Greenhouse gases and dissolved and particulate C*
The main hydrochemical parameters and greenhouse gases concentration and emission fluxes of the
Ket River and its tributaries are listed in **Table 1** and primary data are provided in **Table S2** of the
Supplement. Continuous $pCO_2$ measurements in the main stem during the spring (764 individual data points
over the full distance of the boat route (834 km), demonstrated a lack of systematic change in $CO_2$
concentration from headwaters to the mouth. There were strong but non-systematic variations in $CO_2$





concentrations in the tributaries during the summer (**Fig. 2 A, B**). The $CH_4$ concentration (**Table 1 and Fig.**
**S2 A, B**) was low in the Ket River (around 0.17 and 0.86 µmol $L^{-1}$ in May and August, respectively) and in
the tributaries (range 0.09 to 2.57 µmol $L^{-1}$, 2 to 3 times higher values during the baseflow). These values are
consistent with the range of $CH_4$ concentration in other Siberian Rivers such as Lena (0.03 to 0.199 µmol $L^{-1}$
, Bussman, 2013; Vorobyev et al., 2021). In the Ket River main stem and tributaries, the $CH_4$ concentrations
are 280-1900 and 100-154 times lower than those of $CO_2$ during spring and summer, respectively.
Consequently, diffuse $CH_4$ emissions (**Table 1, Fig. S2 C, D**) constituted 0.1 to 0.5% of total C emissions
and are not discussed in further detail.

During spring flood, $CO_2$ fluxes ranged from 0.26 to 3.2 g C $m^{-2}$ $d^{-1}$ in the main stem and tributaries

(**Table 1; Fig. 2 A**). During baseflow, the flux in the tributaries varied from 0.37 to 7.4 g C $m^{-2}$ $d^{-1}$ and was
a factor of 2 to 3 higher than that in the main stem (**Fig. 2 B, C, Table 1**). Note that peaks of $CO_2$ and $CH_4$
concentration at the main stem were not linked to conflux with tributaries. The $CO_2$ concentration in the river
water and gas transfer velocity assessed from discrete measurements by floating chambers ($K_T$ = 0.08-1.83
m $d^{-1}$ in the main stem; 0.2-1.86 m $d^{-1}$ in the tributaries, **Table 1**) allowed for calculation of the continuous
$CO_2$ fluxes (**Fig. 2 A**). For this, we used an average value of k between two chamber sites (separated by a
distance of 50 to 100 km) to calculate the $FCO_2$ from in-situ measured $pCO_2$ in the river section between
these two sites.

The wind calculated flux was 1.2 to 2 times higher than that measured by chambers, whereas the

calculation with $K_T$ = 4.46 m $d^{-1}$ overestimated the flux by a factor of 3.7 to 6.0. In both cases, the
overestimation of calculated flux relative to chamber-measured flux was most pronounced in the tributaries
rather than in the main stem. Overall, due to small size and short fetch of the Ket River and its tributaries, we
believe that lower values of $K_T$ are more pertinent to the studied river basin. Given that the area is highly
flooded, this is consistent with observations in other flooded regions, where a canopy of vegetation protects
the water-air interface from wind stress thus rendering the gas transfer velocity lower compared to open water
such as large river (i.e., Foster-Martinez and Variano, 2016; Ho et al., 2018; Abril and Borges, 2019). We
therefore warn against the use of high value of transfer velocity, suitable for large rivers of the boreal zone,
for assessing the emissions in medium and small size, sheltered streams with extensive riparian vegetation.





The DIC concentration increased 5 to 10 times between the spring (2.4 to 2.8 mg $L^{-1}$) and summer
baseflow (18 to 20 mg $L^{-1}$) and the pH increased by 0.5-0.7 units between spring freshet and summer baseflow
(**Fig. 3** and **Fig. S3 A, B** of the Supplement). The DOC concentration ranged from 18 to 25 mg $L^{-1}$ during
flood and from 15 to 18 mg $L^{-1}$ during baseflow (**Fig. 3**). There was no systematic variations in DOC
concentration over the 834 km of the main stem (20.7 ± 3.6 and 15.0 ± 1.4 mg $L^{-1}$ in May and August,
respectively); however, it was slightly higher and more variable in the tributaries (22.0± 4.0 and 16.5 ± 7.4
mg $L^{-1}$, **Fig. S3 C, D**). The $SUVA_{254}$ remained highly stable throughout the seasons for both the tributaries
and the main stem (range from 4.2 to 4.9 L mg $C^{-1}$ $m^{-1}$, **Table 1**). The POC was 3 times higher during baseflow
compared to spring and ranged from 2 to 10 mg $L^{-1}$ (**Fig. 3** and **Fig. S3 E, F**). The total bacterial number
ranged from $5.0 \times 10^5$ to $8.7 \times 10^5$ cells $mL^{-1}$ for the main stem and tributaries without significant ($p > 0.05$)
seasonal variation (**Fig. 3** and **S3 G, H**).

*3.2. Diurnal and spatial variation in $CO_2$ concentration and flux*
The diel (day/night) measurements of $CO_2$ concentrations have been performed on six tributaries of
the Ket River during the spring flood period (**Fig. 4**). In two of them (Sochur ad Lopatka) we measured both
$CO_2$ concentration and $CO_2$ fluxes via floating chambers. Continuous $CO_2$ concentrations exhibited a
variation between 5 and 25% of the average value. Only in the case of a small tributary Segondenka (**Fig. 4**
**E**), when we measured $CO_2$ over 38 h, there was a local maximum in concentration between 6 and 7 pm
during the first and second day of monitoring, without any significant link to the water temperature. The
deviation of $FCO_2$ from the average value over the period of observation in two tributaries (**Fig. 4 A, B**) did
not exceed 20%, without any detectable difference between day and night period.
The spatial variation in $pCO_2$ and $FCO_2$ were tested during spring time in the flood zone of the Ket
River middle course, where the flood zone was connected to the main channel. Regardless of the distance
from the main stem and the size of the water body, the variation in $pCO_2$ and chamber-based fluxes were
within 30% of the values measured in the main stem. This suggests that the main stem parameters can be used
for upscaling the C emissions to the overall flood plain during May, provided that the water bodies are
connected to the rivers. Further test of spatial variation were performed on selected small tributaries, when



we moved 8 to 16 km upstream towards the headwaters and monitored the $CO_2$ concentration in the river
water. There was no sizable trend in $CO_2$ concentration over several km length of the tributary, consistent
with small fluctuations over the hundred km-scale of the main stem (**Fig. S4**). Altogether, rather minor spatial
and diel variations in both $CO_2$ concentration and emission fluxes support the chosen sampling strategy and
allow reliable extrapolation of obtained results to full surface of lotic waters of the Ket River basin, during
open water period.

*3.3. Impact of water chemistry and catchment characteristics on $CO_2$ concentration and flux*
There were generally no strong correlations between $CO_2$ and $CH_4$ and the main parameters of the
water column (DOC, DIC, POC, TBC and SUVA (**Table 2**). The $CO_2$ concentration negatively correlated
with $O_2$ concentration ($R_{Pearson} = -0.68$, $p < 0.05$) and $FCO_2$ positively correlated with $SUVA_{254}$ ($R = 0.34$, p
$< 0.05$). Other hydro chemical characteristics of the water column did not impact $CO_2$ and $CH_4$ concentration
and $CO_2$ flux. During spring flood, there was no positive correlation between $FCO_2$ of the river water and
various hydrochemical characteristics. During the summer baseflow, there were positive correlations between
$CO_2$ concentration or flux and SUVA and total bacterial number (**Fig. 5 A, B**).
Among different landscape factors, only deciduous light needleleaf forest (larch trees) exhibited
significant ($p < 0.01$) positive correlations ($0.6 \leq R_S \leq 0.7$) with $CO_2$ concentration and flux of the Ket River
main stem and tributaries, detectable only during the summer baseflow period (**Fig. 5 C**). The peatland and
bogs at the watershed exhibited only weak, although positive ($0.2 < R_S < 0.4$), correlation with $pCO_2$ and
$FCO_2$ (**Fig. 5 D**). The other potentially important landscape factors of the river watershed (type of forest,
riparian and total aboveground vegetation, recent burns, water bodies) as well as lithological parameters
(clays, silts, sands with or without of the presence of carbonate concretions) did not significantly impact the
$CO_2$ and $CH_4$ concentration and measured $CO_2$ fluxes in the Ket River basin (**Table 2**). The mean annual
precipitation (MAP) at the watershed positively correlated with $CO_2$ and $FCO_2$ during the baseflow.







*3.4. Carbon emission and lateral export (yield) of the Ket River basin*

The C emissions (> 99.5 % $CO_2$, < 0.5 % $CH_4$) from the lotic waters of the Ket River basin were assessed based on total river water coverage of the Ket watershed in 2019 (856 km², of which 691 km² is seasonal water, according to the Global SDG database). Given that the measurements were performed at the peak of spring flood in 2019, we used the maximal water coverage of the Ket River basin to calculate the emissions during May and June, and baseflow measurements for July-October period.

For C emission calculation, we used the mean values of $CO_2$ emissions of the main stem and the tributaries ($1.31\pm0.81$ g C m$^{-2}$ d$^{-1}$ for spring flood; $2.11\pm1.86$ g C m$^{-2}$ d$^{-1}$ for summer-autumn baseflow) which covers full variability of both tributaries and the Ket River main channel (**Table 1, Figure 3**). For the month of July which was not sampled in this work and which represents a transition period between the flood and the baseflow, we used the mean value of May and August ($1.55$ g C m$^{-2}$ d$^{-1}$). For the two months of maximal water flow (May - June), the C emission from the whole Ket basin amounts to $68\pm42$ Gg. When summed up with July ($25\pm20$ Gg) and summer-autumn baseflow period (August to October) emission ($32\pm28$ Gg), the total open water season emission flux is 127 Gg. The uncertainty on the total emission over 6 months of the open water period is difficult to quantify but it can be estimated as between 30 and 50 %. This range covers both the uncertainty of the water coverage of the territory and the seasonal and spatial variations of $CO_2$ emission in the Ket basin.

The C export flux (May to October) from the Ket basin was calculated based on monthly-averaged discharge at the river mouth in 2019 available from Russian Hydrological Survey and DOC, DIC and POC concentrations measured in the low reaches of the Ket River in this study (see hydrograph in **Fig. 1**). For this calculation, we used DOC, DIC and POC concentrations measured during spring flood (for May and June period) and baseflow (for August, September and October period). For the month of July, we used the mean concentrations of end of May and August-September which is in accord with seasonal discharge pattern of the Ket River. Note that the contribution of non-studied October month to total open water period water flux is < 10 % and thus cannot provide sizable uncertainties. The total annual (excluding ice-covered period) riverine C export from the Ket River basin ($S_{watershed}$ = 94,000 km²) is 0.35 Tg (3.7 t C km$^{-2}_{land}$ y$^{-1}$), of which DOC, DIC and POC accounts for 56, 24 and 20%, respectively. Therefore, over the 6 month of open water



period, the C emissions from lotic waters of Ket watershed constituted less than 30% of the dissolved and
particulate carbon lateral export from the river basin.


**4. DISCUSSION**
*4.1. Temporal and spatial pattern of $CO_2$ emissions from the river waters*
The first important result of the present study is quite low spatial and seasonal variability in both $CO_2$
concentration and emissions, as well as in DOC concentration and aromaticity (reflected by $SUVA_{254}$) in the
main channel (**Fig. 3, S3, Table 1**). The variability in the tributaries was much larger, with differences in
dissolved and gaseous C parameters between spring flood and summer-autumn baseflow (**Table S3**). While
$CO_2$ concentrations were different between tributaries and the main stem during both flood and baseflow, the
$CO_2$ flux was not different between the main stem and tributaries regardless of season (**Table S4**). This,
together with lack of diel variations in $CO_2$ concentrations and emissions during spring period of maximal
water coverage (**Fig. 4**) suggest rather stable pattern of $CO_2$ in the river water, not linked to short-scale
processes (primary productivity, photolysis, daily temperature variation). Indeed, negligible primary
productivity in the water column may stem from low water temperatures (9.3 °C), shallow photic layer of
organic-rich waters (DOC of 22 mg $L^{-1}$) and lack of periphyton activity during high flow of the spring flood.
Note that this finding contrasts the recent results of high frequency $pCO_2$ measurements in tropical and
temperate world rivers that show a 30 % higher nocturnal emission compared to daytime observations
(Gómez-Gener et al., 2021b).
Concerning spatial variability of C concentrations and emissions during the spring flood, the $pCO_2$
did not demonstrate sizable variation along the main stem of the Ket River and some of its tributaries, when
moving from the mouth to the headwaters.  The SUVA also remained highly stable along the river flow. This,
together with a lack of $pCO_2$ or $FCO_2$ correlation with river watershed area during this period (**Table 2**)
suggest relatively modest control of headwater C cycling by 'fresh' unprocessed organic matter from upland
mire waters. Much stronger control of mire waters is reported in boreal zone of the Northern Europe (Wallin
et al., 2013, 2018). Furthermore, our results on the Ket River main stem and tributaries are in contrast to the





general view of disproportional importance of headwater streams in overall $CO_2$ emission from river basins
(Li et al., 2021). A likely explanation is relative low values of gas transfer velocity measured in the small
streams of the Ket basin in this study (0.2 - 2.0 m d$^{-1}$, **Table 1**). These values are typical of lakes rather than
rivers (i.e., Kokic et al., 2015) and stem from low flow rate, strongly forested and wind-protected river bed
without distinct valley due to generally flat orographic context of this part of the WSL (Serikova et al., 2018).

The second notable result is that, despite sizable variability of $CO_2$ in the tributaries, especially during

the baseflow, there were no correlations between either $pCO_2$ or $FCO_2$ and main hydrochemical parameters
of the water column (**Table 2**). We believe that main reasons of remarkable stability in $CO_2$ concentrations
and emissions and weak environmental control on dissolved and gaseous pattern in the Ket River basin are
(1) essentially homogeneous landscapes, lithology and quaternary deposits of the whole river basin (20-25 %
bogs, 60-70% forest, 3-5 % riparian zone), and (2) strong dominance of allochthonous sources in both
dissolved and particulate organic matter. Indeed, the SUVA and bacterial number (TBC) positively correlated
with both $pCO_2$ and $FCO_2$ during summer (**Fig. 5 A, B**), which may indicate non-negligible role of bacterial
processing of allochthonous (aromatic) DOC delivered to the water column from wetlands and mires.
Furthermore, the positive correlation between mean annual precipitation (MAP) and $pCO_2$ and $FCO_2$ during
the baseflow could reflect the importance of water storage in the mires and wetlands (which also showed
positive but less significant correlations, **Fig. 5 D**) during the summer time, and progressive release of $CO_2$
and DOC-rich waters from the wetlands to the streams. This terrestrial source could be either soil litter
leachates (in spring) or bog water (during baseflow, when the river water is substantially derived from
wetlands, Ala-aho et al., 2018a, b). Although we did not observe correlations between C emission and bog
coverage at the whole Ket River basin, it is known from works in boreal European zone that wetland streams
produce about twice higher $CO_2$ emission flux compared to forest streams (Gomez-Gener et al., 2021). The
patterns in $CO_2$ emissions observed in the present study during summer baseflow thus suggest the importance
of allochthonous organic matter from the peatland for $CO_2$ production in the water column and in soils where
the degradation of DOC is enhanced by the presence of bacteria.

Another interesting correlation is that between $CO_2$ flux during baseflow and the proportion of

deciduous needleleaf forest at the watershed (**Fig. 5 C**), which suggests the importance of C cycling by larch



trees and their possible control on the delivery of degradable organic matter to the river. Similar control of
larch vegetation on riverine $CO_2$ has been suggested for the Lena River, Eastern Siberia (Vorobyev et al.,
2021) although we acknowledge that further observations on contrasted Siberian watersheds are necessary to
confirm the observation that larch trees litterfall led to export of degradable OM to the river.

During both spring flood and summer baseflow, punctual local variations in $CO_2$ concentration and

emissions along the sampling route of the main stem (**Fig. 2 A**) were not necessarily linked to $CO_2$-rich
tributaries, or variations in water chemistry of the specific segments of the river. Similar to other studies of
boreal and subarctic rivers (i.e., Vorobyev et al., 2021; Lundin et al. 2013, Rocher-Ros et al. 2019), these
variations likely reflect local processes in the main stem, such as lateral influx from the shores and shallow
subsurface waters, sediment resuspension and respiration, or the discharge of underground, $CO_2$-rich fluids
in the river bed (hyporheic zone). Thus, via comprehensive analysis of 187 streams and rivers across the
contiguous United States, Hotchkiss et al. (2015) demonstrated that ~60% of $CO_2$ evasion is from external
sources rather than internal production. In view of lack of correlation of $CO_2$ emissions in the Ket River and
tributaries with hydrochemical parameters of the water column, we believe that external source of $CO_2$ in
studied river system represents sizable contribution to total riverine $CO_2$ evasion across the seasons and
sampling sites. In particular, in small peatland streams, the $CO_2$-rich deep peat/groundwater is known to be
the major source of aquatic $CO_2$ under low flow conditions (Dinsmore and Billett, 2008), whereas in boreal
headwater streams of N Sweden the main source of stream $CO_2$ was inflowing $CO_2$-rich soil waters
(Winterdahl et al., 2016).

At the Ket River basin, the local soil/groundwater effects are certainly more pronounced during

baseflow, due to lower impact of dilution, compared to the spring flood period. The hypothesis of deeper flow
path in summer compared to spring is confirmed for the WSL (Frey and McClelland, 2009; Pokrovsky et al.,
2015; Serikova et al., 2018) and is supported in this study by a strong increase in DIC concentration between
spring and summer (**Fig. 3**). Thus, although the pairwise correlations between parameters do not support any
particular mechanism, it is not excluded that OM bio- and photo degradation and local mire water feeding
drive $FCO_2$ in spring, and that deeper flowpaths and DIC export drive the elevated $FCO_2$ in summer.



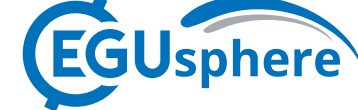

Another important factor responsible for higher $CO_2$ production in the water column in summer
compared to spring could be POC degradation. The riverine POC is known to be more biodegradable than
DOC (Attermeyer et al., 2018), and the POC concentration in the Ket River basin increased 4-fold between
spring and summer (**Table 1**). The origin of summer-time POC and its lability remain elusive, but could be a
combination of plankton bloom and mire- or forest-derived DOC coagulation products in the water column
(Krickov et al., 2018). Furthermore, pronounced heterogeneity in $CO_2$ emission during baseflow among
tributaries may also reflect the heterogeneity of riverine organic matter which is known to be the maximal
during low flow conditions and minimal during high flow (Lynch et al., 2019).
Taken together, the present study demonstrates rather stable and non-equilibrium behavior of $CO_2$ in
the Ket River basin, with minimal role of *hot spots* from various local sources. In this regard, we note high
representability of studied riverine system for large pristine zones of taiga forest and bog regions of the WSL
- eastern smaller tributaries of the Ob River in permafrost -free zone (Chulym, Tym, Vakh, Agan, Trom'egan),
and also western tributaries of the Yenisey River (Dubches, Sym and Kas) with total watershed area of
350,000 km². To which degree the Ket River can serve as an analogue of another eastern tributary of the Ob
River, the more anthropogenically and agriculturally - impacted tributary Chulym River ($S_{watershed} = 134,000$
km²), remains unknown.

*4.2. Emissions from the Ket River basin compared to lateral export of riverine carbon*
The estimated C emissions (> 99.5 % C; < 0.5 % $CH_4$) from the Ket River main channel over 830 km
distance (0.5 to 2.5 g C m$^{-2}$ d$^{-1}$) are comparable to those of the Kolyma River (0.35 g C m$^{-2}$ d$^{-1}$ in the main
stem and 2.1 g C m$^{-2}$ d$^{-1}$ for lotic waters of the basin; Denfeld et al., 2013), the Ob River main channel
(1.32±0.14  g C m$^{-2}$ d$^{-1}$ in the permafrost-free zone; Karlsson et al., 2021), and the Lena River (0.8 to 1.7 g C
m$^{-2}$ d$^{-1}$; Vorobyev et al., 2021). The $CO_2$ emission in Ket's tributaries (1 to 2 g C m$^{-2}$ d$^{-1}$ in spring; 1 to 5 g C
m$^{-2}$ d$^{-1}$ in summer) are within the range reported for small rivers and streams of the permafrost-free zone of
western Siberia (0 to 3.6 g C m$^{-2}$ d$^{-1}$ in spring; 4 to 9 g C m$^{-2}$ d$^{-1}$ in summer; Serikova et al., 2018), forest and
wetland headwater streams of northern Sweden (0.5 to 5 g C m$^{-2}$ d$^{-1}$; Gomez-Gener et al., 2021), rivers and
headwater streams of the Unites States (2.7 to 3.1 g C m$^{-2}$ d$^{-1}$, Butman and Raymond, 2011; Hotchkiss et al.,



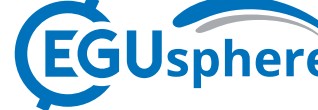

2015), small mountain streams in Northern Europe (3.3 g C m$^{-2}$ d$^{-1}$, Rocher-Ros et al., 2019), boreal streams
in Canada and Alaska (0.8 to 5.2 g C m$^{-2}$ d$^{-1}$, Koprivnjak et al., 2010; Teodoru et al., 2009; Crawford et al.,
2013; Campeau et al., 2014).
Total C emissions from the water surfaces of the Ket River basin assessed in this study (148 g C-CO$_2$
m$^{-2}$ y$^{-1}$, assuming no emission under ice) are lower than those of the lotic waters of western Siberia (898 g C-
CO$_2$ m$^{-2}$ y$^{-1}$, Karlsson et al., 2021) but comparable to global C emissions from the Lena river basin (180 to
360 g C m$^{-2}$ y$^{-1}$, Vorobyev et al., 2021). When normalized to the Ket river basin area (S$_{watershed}$ = 94,000 km²),
the C emission amounts to 1.35 g C m$^{-2}_{land}$ y$^{-1}$. Hutchins et al. (2020) reported 0.63 to 0.29 g C-CO$_2$ m$^{-2}_{land}$ y$^{-}$
$^{1}$ emission from 50 small streams in boreal biome of Canada, comparable to the headwater stream network
emissions in Alaska (0.44 g C m$^{-2}$ y$^{-1}$, Crawford et al., 2013) and Zolkos et al. (2019) found approximately
0.4 g C m$^{-2}$ y$^{-1}$ in the Northwest Territories. Much higher land area - specific emissions, comparable or
exceeding those of the Ket River, were reported in Québec (1.0 to 4.6 g C m$^{-2}$ y$^{-1}$; Campeau and del Giorgio,
2014; Hutchins et al., 2019; Teodoru et al., 2009), Sweden (1.6 to 8.6 g C m$^{-2}$ y$^{-1}$; Humborg et al., 2010;
Jonsson et al., 2007; Lundin et al., 2013; Wallin et al., 2011, 2018) and boreal portions of the Yukon River
(7 to 9 g C m$^{-2}$ y$^{-1}$; Striegl et al., 2012; Stackpoole et al., 2017). Possible reasons for these differences could
be different areal coverage of the territory by river network, the calculated rather than measured CO$_2$ fluxes,
or the higher gas transfer velocity in the rivers from mountainous regions.
The regional assessment of the Ket River basin performed in this study are based on direct chamber
measurements of emissions and as such provide rigorous basis for upscaling the CO$_2$ emissions from currently
understudied lotic waters of permafrost-free zone of Western Siberia. The C evasion from the Ket basin
assessed in the present work (127 ± 11 Gg y$^{-1}$, ignoring the emission during the ice breakup in early spring)
is 3 times lower than the total (DOC+DIC+POC) lateral export by this river from the same territory (0.35 Tg
C y$^{-1}$ ). The lateral C loss (yield) for the Ket River (3.7 t C km$^{-2}_{land}$ y$^{-1}$) is in agreement with regional C
(DOC+DIC) yield by permafrost-free small and medium size rivers of the WSL (3 to 4 t C km$^{-2}_{land}$ y$^{-1}$,
Pokrovsky et al., 2020) and with the Ob River in its the middle course at the latitude of the Ket River (3.6 t
C km$^{-2}_{land}$ y$^{-1}$, Vorobyev et al., 2019). Such high C yields in the southern, permafrost-free part of the WSL
stem from essentially inorganic carbon originated from groundwater discharge of carbonate mineral rich



reservoirs, abundant in this region (Pokrovsky et al., 2015). At the same time, the organic C yield in rivers of
this region is quite low and represents less than 20% of total C yield (Pokrovsky et al., 2020; Vorobyev et al.,
2019). This can explain anomalously low value of C evasion : C export of the Ket River (1 : 3) measured in
this work as compared to the average values for permafrost-free zone of Western Siberia (1 : 1, Serikova et
al., 2019). One should also note that the gas transfer velocity measured in thus study provides much lower
fluxes than those calculated with $K_T = 4.46$ m d$^{-1}$ in previous studies (**Table S2**). Another factors potentially
leading to underestimation of C evasion in this study is GIS-based minimal water coverage which does not
include seasonal oxbow lakes, flooded forest and temporary water bodies of the floodplain which provide
sizable emissions (see Krickov et al., 2021). We also do not exclude that some important hot moments / hot
spots of C emission were missed in our sampling campaign, such as summer baseflow/autumn peaks
(Serikova et al., 2019) or stagnant zones of the floodplain in summer (Krickov et al., 2021; Castro-Morales
et al., 2021). This calls a need for higher spatial and temporal resolution monitoring of C emission, with
special focus on important events across full hydrological continuum.

**5. Concluding remarks**

Via combination of discrete floating chamber and hydrochemistry and continuous $CO_2$ concentration

measurements over 830 km of large pristine boreal river of western Siberia main channel and its 26 tributaries
during the peak of spring flood and the summer-autumn baseflow, we quantified spatial and temporal
variations, overall emissions of C ($CO_2$, $CH_4$) and export of (DOC, DIC and POC) during the 6 months of
open water period.  The range of $CO_2$ and $CH_4$ concentrations in the main channel and tributaries as well as
$CO_2$ emissions were consistent with other boreal and subarctic regions but demonstrated rather low seasonal
and spatial variability. The diel $CO_2$ flux by floating chambers and continuous $pCO_2$ measurements in the
tributaries of the Ket River during spring flood demonstrated negligible impact of day/night period on the
$CO_2$ concentrations and emission fluxes. During spring flood, there were no correlations between
concentrations of $CO_2$ and $CH_4$, or $CO_2$ flux and their main potential controlling physiochemical parameters
of the water column  as well as climatic and landscape parameters of the watershed.




We hypothesize that homogeneous landscape coverage (bog and taiga forest) provide stable
allochthonous input of DOM as confirmed by very weak spatial and seasonal variations of DOM aromaticity.
Among possible driving factors of $CO_2$ production in the water column (bio- and photo-degradation of DOC
and POC, plankton metabolism), none seems to be sizably important for persistent $CO_2$ supersaturation and
relevant emissions. The landscape factors of the watershed (bog and forest coverage, soil organic carbon
stock) of the tributaries and along the main stem did not sizably affected the C concentration and emission
pattern across two seasons. We hypothesize that stable terrestrial input of strongly aromatic DOM, shallow
photic layer and humic waters of the Ket River basin preclude sizable daily and seasonal variations of C
parameters. Punctual discharge of groundwaters, resuspension of sediments or shallow subsurface influx from
mires and riparian zone may be responsible for small-scale heterogeneities in C emissions and concentrations
along the main stem and among the tributaries. These effects are much stronger pronounced during summer
baseflow compared to spring flood. Overall, deeper flow paths in summer compared to spring enhance the
DIC discharge within the river bed and the tributaries, thus leading to elevated $CO_2$ flux in summer.
Additional factor responsible for higher $CO_2$ emission during this season could be mire-originated particulate
organic matter (POM) processing in the water column. Further experiments on POM degradation and isotope
tracing of C sources are therefore needed to quantitatively discriminate between surficial "organic" and deep
"inorganic" source of $CO_2$ in the Ket River basin during summer baseflow. In this regard, a reason for
relatively low spatial and temporal variability of $CO_2$ concentration and emissions in this large river basin
could be that existing variations in C supply and control of $FCO_2$ are coupled and counteract each other so
that the net $FCO_2$ remains spatially and temporally stable.
The six month open-water period C emissions from the lotic waters of the Ket River basin were sizably
lower than the lateral C export by this river during the same period.  We conclude that regional estimations
of C balance in lotic systems should be based on a combination of direct chamber measurements, discrete
hydrochemical sampling and continuous in-situ monitoring with submersible sensors, at least during two most
important hydrological periods of the year which are, for boreal regions, the spring flood and the summer-
autumn baseflow. We believe that this is the best trade-off between scientific rigor and logistical feasibility
in poorly accessible, pristine and strongly understudied regions.




**Acknowledgements.**
We acknowledge support from RSF grant 22-17-00253, RFBR grant 20-05-00729, the TSU Development
Program "Priority-2030", grant "Kolmogorov" of MES (Agreement No 075-15-2022-241), and the Swedish
Research Council (grant no. 2016-05275).

**Authors contribution.**
AL and OP designed the study and wrote the paper; AL, SV, IK and OP performed sampling, analysis and
their interpretation; LS performed bacterial assessment and DOC/DIC analysis and interpretation; MK
performed landscape characterization of the Ket River basin and calculated water surface area; SK
performed hydrological analysis; JK provided analyses of literature data, transfer coefficients for $FCO_2$
calculations and global estimations of areal emission vs export.

**Competing interests.**
The authors declare that they have no conflict of interest.

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




**Table 1.** Measured hydrochemical and GHG exchange parameters in the Ket River main stem and
tributaries (average ± s.d.; (*n*) is number of measurements).

| Parameter | unit | Tributaries | | Main stem | |
|---|---|---|---|---|---|
| | | Flood (*n=26*) | Base flow (*n=12*) | Flood (*n=7*) | Base flow (*n=6*) |
| Water temperature | °C | 9.48±2.25 | 14.9±1.24 | 9.06±1.59 | 16.5±0.54 |
| pH | | 6.31±0.45 | 6.71±0.57 | 6.2±0.43 | 7.29±0.26 |
| Dissolved $O_2$ | mg L$^{-1}$ | 8.53±1.26 | 8.02±1.13 | 8.85±0.83 | 8.78±0.18 |
| Specific Conductivity | μS cm$^{-1}$ | 40.7±22.7 | 126.9±62.1 | 39±14.9 | 181±36.8 |
| DIC | mg L$^{-1}$ | 2.83±2.58 | 17.8±10.4 | 2.43±1.49 | 20.5±5.22 |
| DOC | mg L$^{-1}$ | 21.7±3.94 | 15.7±7.04 | 21.9±4.28 | 16.6±3.57 |
| SUVA$_{254}$ | L mg C$^{-1}$ m$^{-1}$ | 4.34±0.33 | 4.9±0.66 | 4.29±0.18 | 4.26±0.52 |
| PON | mg L$^{-1}$ | 0.08±0.06 | 0.64±0.27 | 0.1±0.07 | 0.96±0.22 |
| POC | mg L$^{-1}$ | 2.41±1.17 | 8±2.36 | 2.55±1.2 | 9.49±1.98 |
| TBC | *10$^5$ cells ml$^{-1}$ | 5.89±3.26 | 8.69±3.21 | 5.95±2.83 | 4.94±2.15 |
| $K_T$ | m d$^{-1}$ | 0.53±0.38 | 1.21±0.52 | 0.77±0.55 | 1.22±0.37 |
| FCO$_2$ | g C m$^{-2}$ d$^{-1}$ | 1.3±0.76 | 2.63±2.15 | 1.35±1.08 | 1.16±0.5 |
| pCO$_2$ | μatm | 2877±679 | 4005±1494 | 2405±328 | 2523±981 |
| FCH$_4$ | mmol C m$^{-2}$ d$^{-1}$ | 0.39±0.95 | 1.38±1.21 | 0.06±0.05 | 0.95±0.88 |
| CH$_4$ | μmol L$^{-1}$ | 0.65±0.66 | 1.17±0.81 | 0.17±0.01 | 0.86±0.91 |





**Table 2.** Pearson correlation coefficients of measured $FCO_2$, $CO_2$, and $CH_4$ concentration with hydrochemical parameters of the water column (DOC, SUVA, particulate organic carbon and nitrogen, total bacterial cells) and landscape parameters of the tributaries and the main stem of the Ket River. Significant ($p < 0.05$) values are labeled by asterisk.

| | all seasons | | | flood | | | baseflow | | |
|---|---|---|---|---|---|---|---|---|---|
| | $CH_4$ | $CO_2$ | $FCO_2$ | $CH_4$ | $CO_2$ | $FCO_2$ | $CH_4$ | $CO_2$ | $FCO_2$ |
| **Hydrochemical parameters** | | | | | | | | | |
| pH | 0.2 | -0.1 | -0.2 | -0.1 | 0.1 | -0.2 | 0.0 | -0.6* | -0.6* |
| Dissolved $O_2$ | -0.1 | -0.7* | -0.1 | 0.0 | -0.8* | 0.1 | -0.2 | -0.8* | -0.7* |
| Specific conductivity | 0.3 | 0.0 | 0.1 | -0.2 | 0.0 | 0.1 | 0.2 | -0.3 | -0.6* |
| DIC | 0.3 | 0.0 | 0.0 | -0.1 | 0.0 | 0.1 | 0.2 | -0.4 | -0.7* |
| DOC | -0.1 | 0.0 | 0.1 | 0.3 | 0.0 | -0.1 | -0.2 | -0.1 | 0.2 |
| SUVA$_{254}$ | 0.1 | 0.2 | 0.3 | 0.4 | -0.3 | 0.1 | -0.2 | 0.5* | 0.6* |
| PON | 0.1 | -0.1 | 0.2 | -0.2 | -0.4* | 0.2 | -0.4 | -0.5* | -0.5 |
| POC | 0.1 | -0.1 | 0.2 | -0.2 | -0.4* | 0.1 | -0.3 | -0.3 | 0.1 |
| TBC | 0.2 | 0.2 | 0.1 | 0.3 | -0.2 | -0.1 | 0.0 | 0.5* | 0.5* |
| **Climatic characteristics** | | | | | | | | | |
| MAAT | 0.2 | 0.0 | -0.5* | 0.1 | 0.0 | -0.4* | 0.2 | 0.1 | -0.5 |
| MAP | 0.0 | 0.3* | 0.5* | 0.1 | 0.0 | 0.3 | 0.1 | 0.6* | 0.7* |
| **Land-cover characteristics** | | | | | | | | | |
| Watershed area | -0.3 | -0.3* | 0.2 | -0.4 | -0.5* | 0.0 | -0.2 | -0.1 | 0.5 |
| Dark Needleleaf Forest | 0.1 | 0.0 | -0.3 | 0.1 | 0.0 | -0.3 | 0.2 | -0.1 | -0.2 |
| Light Needleleaf Forest | 0.3* | 0.4* | 0.2 | 0.4 | 0.2 | 0.0 | 0.4 | 0.7* | 0.6* |
| Broadleaf Forest | -0.3 | -0.4* | 0.1 | -0.5* | -0.4 | 0.1 | -0.3 | -0.6* | -0.2 |
| Mixed Forest | 0.0 | -0.2 | -0.3 | 0.1 | -0.1 | -0.3 | -0.1 | -0.4 | -0.4 |
| Peatlands and bogs | 0.0 | 0.2 | 0.3 | -0.1 | 0.0 | 0.2 | 0.1 | 0.2 | 0.4 |
| Riparian Vegetation | -0.1 | 0.0 | -0.1 | -0.2 | 0.1 | 0.0 | -0.2 | -0.2 | -0.5 |
| Grassland | 0.1 | -0.1 | 0.0 | -0.1 | -0.2 | 0.1 | 0.3 | 0.0 | -0.5 |
| Recent Burns | -0.1 | -0.1 | 0.2 | -0.1 | -0.2 | 0.1 | -0.3 | 0.1 | 0.4 |
| Water Bodies | -0.2 | -0.1 | 0.3 | -0.3 | -0.3 | 0.2 | -0.2 | -0.1 | 0.3 |
| **Lithology characteristics** | | | | | | | | | |
| Upper Cretaceous, Maastrichtian – (sedimentary, silicate) Lower Paleocene (sedimentary silicate rocks) | 0.1 | -0.4* | 0.0 | 0.3 | -0.3 | 0.2 | 0.0 | -0.5* | -0.4 |
| Paleogene. Upper Oligocene (clays and silts) | 0.1 | -0.2 | 0.1 | 0.1 | -0.1 | 0.2 | 0.0 | -0.5* | -0.2 |
| Cretaceous.Coniacian – Campanian (carbonates) | -0.2 | -0.4* | -0.3 | -0.2 | -0.2 | -0.2 | -0.3 | -0.7* | -0.6* |
| Neogene. Lower -Middle Miocene (clays, silts) | -0.1 | 0.2 | 0.3 | -0.1 | 0.0 | 0.2 | -0.1 | 0.3 | 0.3 |
| Upper Pliocene-Eopleistocene (sands) | 0.0 | 0.2 | 0.1 | 0.0 | 0.2 | 0.0 | 0.0 | 0.3 | 0.3 |
| Cretaceous.Cenoman – Turon (clays, some carbonates) | -0.2 | -0.5* | -0.3 | -0.3 | -0.3 | -0.2 | -0.3 | -0.7* | -0.6* |
| Neogene. Lower Miocene (sands) | 0.1 | -0.2 | -0.1 | | -0.2 | -0.2 | 0.1 | -0.3 | 0.0 |



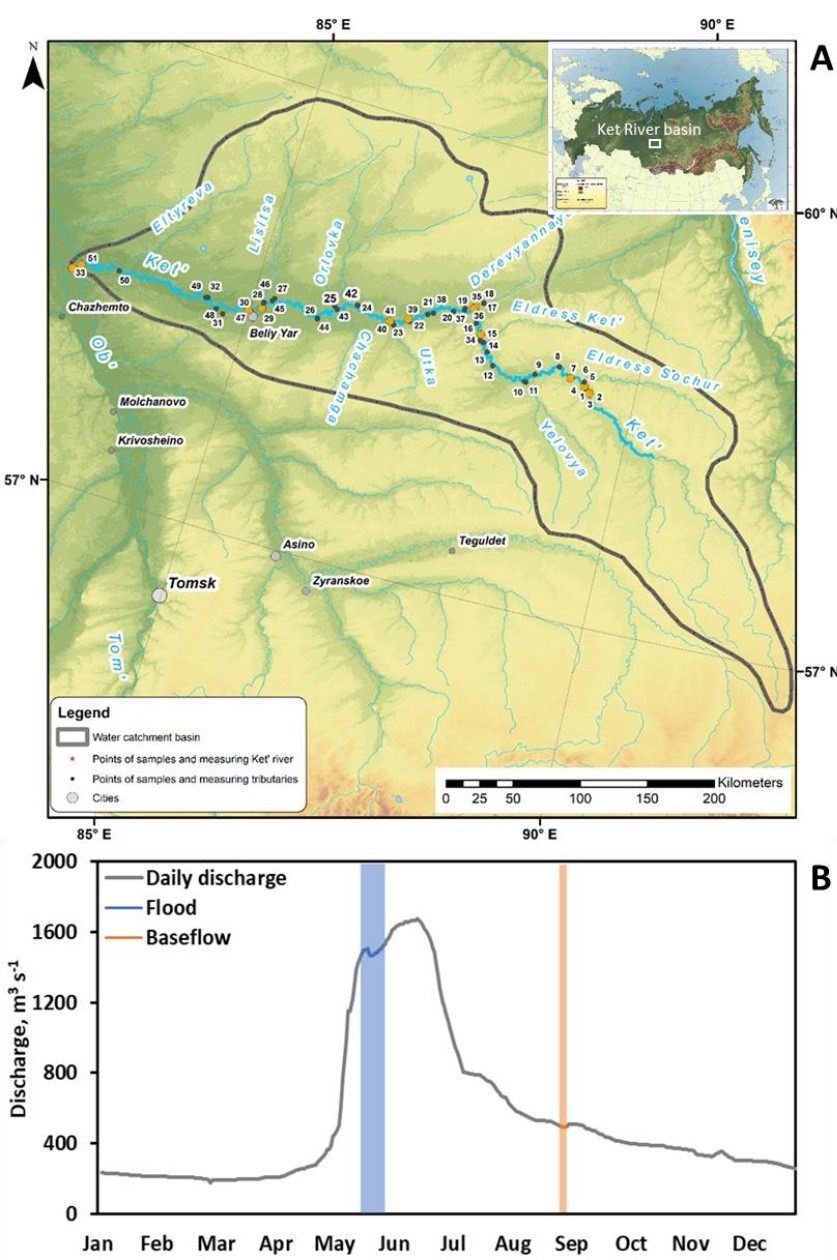

**Fig. 1. A:** Map of the studied Ket River watershed with continuous pCO₂ measurements in the main stem. **B:** Daily discharge (Q) at the gauging station of the Ket mouth, Rodionovka, in 2019. Highlighted in blue and orange are two sampling campaigns of this study, spring flood and summer-autumn baseflow.





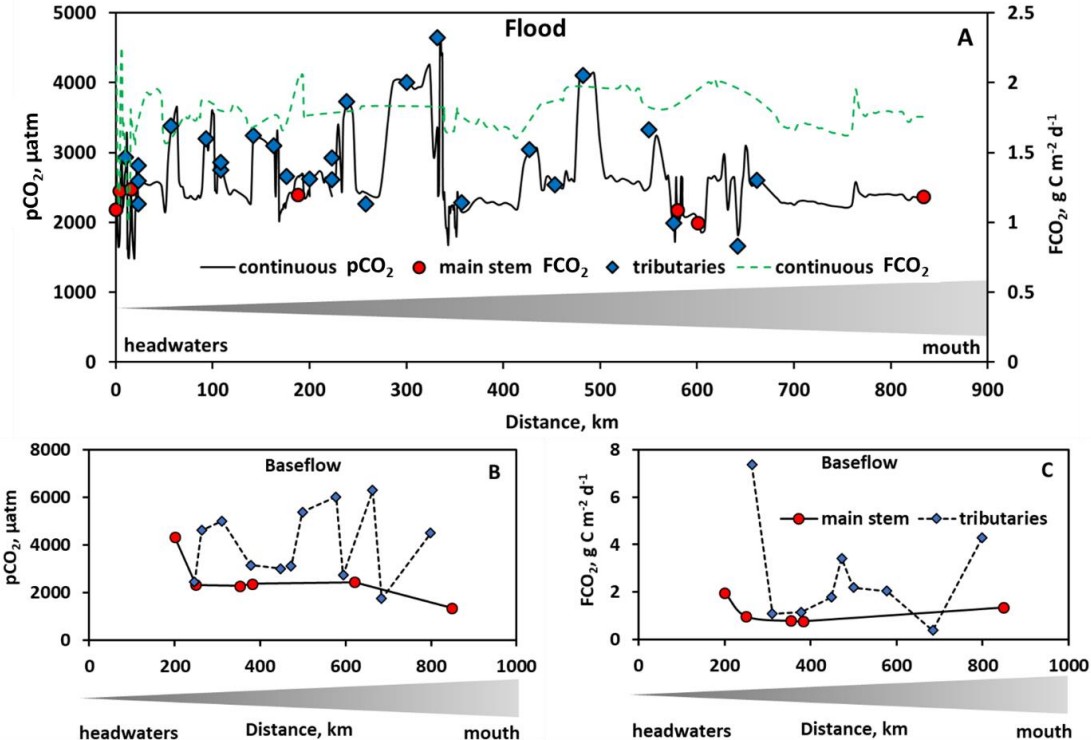

**Figure 2.** The $pCO_2$ and measured and calculated $CO_2$ fluxes during spring (**A**) and summer (**B, C**) of the Ket River main stem and tributaries (over the 830 km distance, from the headwaters to the mouth (left to right). Continuous $CO_2$ measurements in (**A**) are only for the main stem. Note that during summer baseflow, the water level did not allow reaching the headwaters of the Ket River (first 0-200 km on the river course).







875

876

877

**Figure 3.** Mean (± s.d.) GHG concentration and fluxes, hydrochemical parameters, particulate organic carbon and nitrogen (POC and PON, respectively) and total bacteria count (TBC), in the main channel (orange column) and the tributaries (blue column) of the Ket River in spring flood and summer (early fall) baseflow.








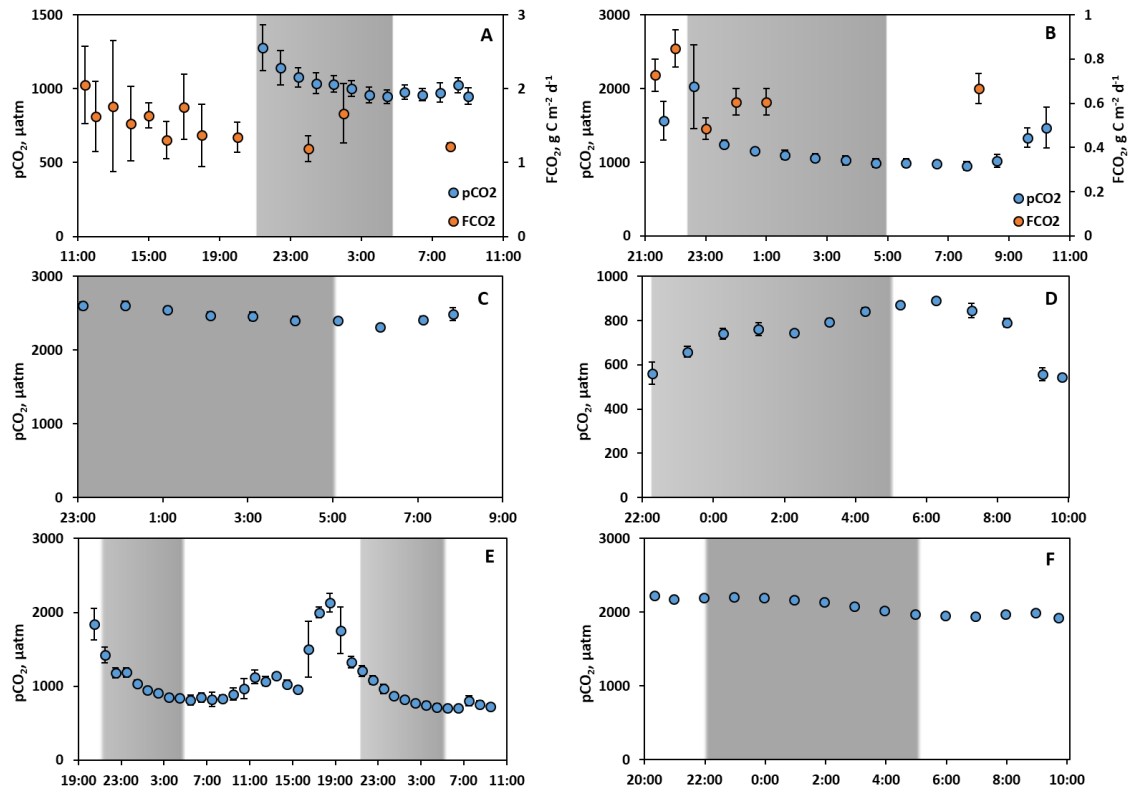




**Figure 4.** Continuous $pCO_2$ concentration (**A-F**, blue circles) and chamber-based fluxes (**A, B**) measured during spring flood period in tributaries (**A** Sochur No 3, **B** Lopatka No 8, **C** Derevyannaya No 12, **D** Ob river entrance, **E** Segondenka No 26) and in the Ket River main stem (middle course) near Stepanovka village (**F**) including night time measurements (shaded area). Variations of water temperature were within the range of 0.3 to 0.6 °C and did not exhibit significant correlations with $pCO_2$ and $FCO_2$.











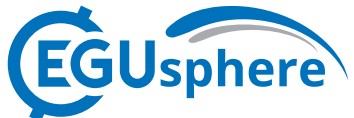


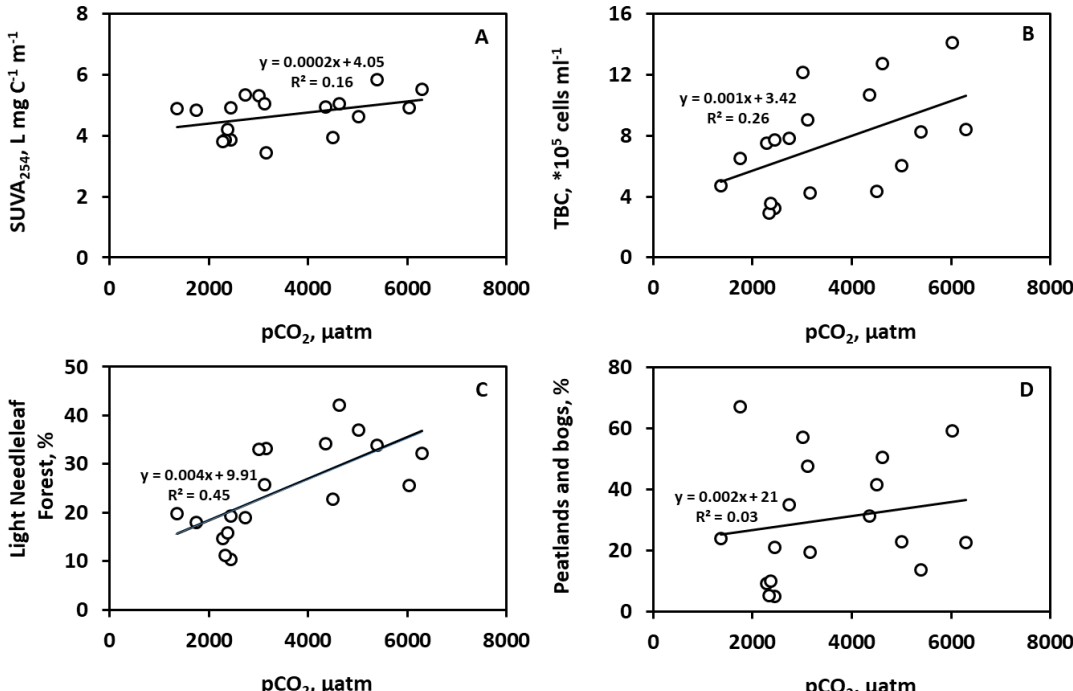


**Figure 5.** Significant ($p < 0.05$) positive control of SUVA (**A**), Total Bacterial Count (**B**), Light needleleaf forest (**C**) and wetlands (**D**) on $CO_2$ concentration in the Ket River and tributaries during summer baseflow.
