# Peer review of "Carbon emission and export from Ket River, western Siberia"

_EGUsphere, 2022_

## Author Comment (AC1)

**General comment**
Reviewer: In this work, Lim and colleagues reported the spatial and seasonal dynamics of C export and emissions from the Ket River mainstem and major tributaries by combining continuous in-situ measurements and discrete sampling. Although high latitude regions are an important component of the global carbon cycle due to their large carbon stocks, carbon emissions and export from permafrost-affected regions, especially those in Russia, are poorly studied due to logical constraints and inaccessibility. In view of the changing climate and thawing permafrost, this study is timely important in quantitatively assessing the spatial and seasonal patterns of dissolved carbon export and emissions in this permafrost-affected river basin and thus provides important insights into future riverine carbon cycling. This research work fits well with the scope of the journal Biogeosciences. But there are several major issues to be properly addressed during the revision stage.
**Response: We are grateful to generally positive evaluation of our work and greatly revised the manuscript following the reviewer's comments.**

Reviewer: My first major comment is on the observed stable behavior of CO2 in the Ket River basin. The authors have tried to explain the stable behavior of the CO2 dynamics (pCO2 and Fco2) by relating them to various physiochemical parameters. But it seems none of the physiochemical parameters is sufficiently strong to drive the pattern although they show pronounced spatial and seasonal variations, as shown in Table 1 and Figs 2 and 3. This is contrary to studies in other climates/regions. I am wondering whether these potential drivers are working in different (opposing) directions and have counteracted each other. The authors may need to think about this seriously, and re-examine the cause-effect relationships. Many of the current discussion statements are lack of evidence and speculative.
**Response: We basically agree with this remark: none of the studied physico-chemical or landscape parameters is capable explaining the observed pattern. To test the possibility suggested by the reviewer – that potential drivers are working in different (opposing) directions and have counteracted each other – we performed a multi-parameter statistics of the full data set (Table S2 together with land cover parameters of the watershed) via PCA, but this did not allow identifying the main drivers and actually, the overall explanation capacity of two factors was below 26%. In addition to PCA, a Redundancy Analysis (RDA) was used to extract and summarize the variation in a set of response variables in C pattern that can be explained by a set of explanatory variables (environmental, climatic and hydrochemical factors). The RDA treatment did not provide additional insights into environmental control of C pattern across the rivers and seasons. After normalization, the main result was that the analyses are not statistically significant ($p > 0.05$).**
**However, we agree with the reviewer that the possibility of governing factors that counteract each other cannot be excluded, and we added pertinent sentences to the revised text of the Discussion (end of section 4.1). Given that even a multiparametric statistics (PCA) did no demonstrate sizable explanation capacity of the data set, we cannot exclude that these potential physico-chemical, microbiological and landscape drivers are working in different (opposing) directions and have counteracted each other. However, further in-depth analysis of these interactions require much better seasonal resolution ideally over full period of the year, which was beyond the scope of the present study.**
**Note that a more likely explanation of remarkable stability in $CO_2$ concentrations and emissions and weak environmental control on dissolved and gaseous pattern in the Ket**

**River basin are (1) essentially homogeneous landscapes, lithology and quaternary deposits of the whole river basin (20-25 % bogs, 60-70% forest, 3-5 % riparian zone), and (2) strong dominance of allochthonous sources in both dissolved and particulate organic matter. The latter is consistent with the finding that the SUVA and bacterial number (TBC) positively correlated with both $pCO_2$ and $FCO_2$ during summer (Fig. 5 A, B), which may indicate non-negligible role of bacterial processing of allochthonous (aromatic) DOC delivered to the water column from wetlands and mires. Furthermore, the positive correlation between mean annual precipitation (MAP) and $pCO_2$ and $FCO_2$ during the baseflow could reflect the importance of water storage in the mires and wetlands (Fig. 5 D) during the summer time, and progressive release of $CO_2$ and DOC-rich waters from the wetlands to the streams.**

**To summarize our response, we did identify some possible drivers (allochthonous DOM, bacteria, mire coverage) but these factors operated essentially during the summer baseflow and could not explain the full spatial and seasonal pattern of C concentration and emission in the Ket River basin.**

Reviewer: My second major comment is on the calculation of the annual flux of $CO_2$ emission and lateral C export. With very limited C sampling results covering a short period (Fig 1b), the annual flux estimates are prone to large errors. For example, $CO_2$ emissions during ice melting periods are exceptionally strong after a long period of $CO_2$ accumulation. But such emissions are not included or accounted for in the estimation.

**Response: We understand and partially agree with this concern of the reviewer. However, our main argument that via performing both peak of the spring flood and summer baseflow sampling campaigns with unprecedented spatial resolution, we encompassed most important open-water period for the $CO_2$ and $CH_4$ evasion. In this regard, we believe that the spring flood (May-June) and summer baseflow (July-August-September) are largely sufficient to represent the majority of C evasion from the river waters. In fact, similar to previous study of rivers along a 2500 km transect of the WSL territory, the timing of the two sampling campaigns covered approximately 80% of the annual water discharge in the basins (Serikova et al., 2018). However, to better argument our response, in the figure below (Fig. R1) we presented unpublished data of our group on one site of the main stem and several small tributaries of the Ket River sampled in spring, summer and autumn (October before ice-on). It can be seen that the $CO_2$ concentration and emission flux during October are either equal or 1.5-2 times lower than that during summer (August). Because we postulated that the evasion during autumn is equal to that during summer baseflow, the assessment of overall C evasion from the Ket River basin used in the present study cannot sizably underestimate the real values.**

[Figure]

**Fig. R1.** A histogram of $pCO_2$ (A) and $FCO_2$ (B) in the Ket River main stem and several tributaries during three main open-water seasons. Unpublished data of our group.

**Further, we explicitly stated that the study is focused on six months of open water period and we could not investigate the winter-time (under-ice) accumulation of GHG or a number of logistical reasons: one would not risk remaining on the river ice to capture the gas regime during ice cracking when the river physically 'explodes'. To the best of our knowledge, none of the Siberian river has been sampled for winter time C evasion so far, and this clearly requires a study in its own. Doing this on a small tributary could be an option, and this research is in progress by our group.**

Reviewer: Likewise, the lateral fluxes based on monthly average discharge are likely with huge uncertainty. E.g., the strong DIC concentration differences between the flood and baseflow (Table 2) suggest significant dilution effect and changing flow paths.
**Response: This is very pertinent remark. Our main argument on the validity of dissolved C (DOC and DIC) export fluxes used in the present study is a similarity of the total C yield for the Ket River (3.7 t C km$^{-2}_{land}$ y$^{-1}$) and 1) values of the regional C (DOC+DIC) yield by permafrost-free small and medium size rivers of the WSL (3 to 4 t C km$^{-2}_{land}$ y$^{-1}$, Pokrovsky et al., 2020) and 2) the Ob River in its the middle course (3.6 t C km$^{-2}_{land}$ y$^{-1}$, Vorobyev et al., 2019). These former studies of our group were performed with much better seasonal resolution, including both open water and glacial period of the year. For example, the latter study of the Ob River, which is very similar in the environmental context to the Ket River, actually included high frequency weekly sampling over several years of monitoring.**

Reviewer: Overall, this manuscript was well written, but the structure could be further improved by moving the discussion statements from the Results section to the Discussion section.
**Response: We followed the recommendations of the reviewer and revised the manuscript accordingly.**

A further language editing is also needed before its resubmission.
**Response: We carefully check for spelling and grammar errors and improved the style of many sentences in the revised version. Note that the APC of accepted manuscripts include full English proofread of the text.**

Specific comments (with line number):
L42-43: 100 to 150 times?
**Response: Yes, revised accordingly**

L64: even for these regions, the estimates are still with great uncertainty.
**We agree and alerted the reader about uncertainty on these estimates**

L80: delete 'remain'
**Response: Revised accordingly**

L95: essentially speaking, the two sampling campaigns represent the two extremes (highest flow and lowest flow, respectively). A question then is whether it is reasonable to use these extremes for annual flux estimation (emission and downstream export)?
**Response: Please note that a combination of natural factors such as low runoff, lack of relief and highly homogenous landscape coverage of the permafrost-free zone of western Siberia in general and of the Ket River basin in particular provides quite smooth hydrographs of the rivers. In this regard, the spring flood period is extended over 2 month, from the beginning of May to middle of July, whereas summer baseflow includes second half of July, August and September. We added this information in the revised text (section 2.1).**

L108: what is hydrocarbon exploration? I don't understand this.
**Response: This means that there is no oil and gas development and production activity in the Ket watershed area, revised the text.**

L113: delete '.' after -0.6. also, references are needed to this paragraph describing the background information.
**Response: Revised accordingly and added some references (Frey and Smith, 2007; Pokrovsky et al., 2015) as requested.**

L119: Have the authors finished the cruise (1300 km in total) and sampling within 3 days? Sounds an impossible task.
**Response: There were two boat trips in this study. The spring time cruise took 11 days on the river for 1309 km overall trip length. During summer baseflow, the 4-days trip was shortened by 200 km due to too low water level in the headwaters and some tributaries. We added this missing information in the revised text. Note that we did not perform day/night monitoring in August which allowed greatly shortening the overall cruise time.**

L125-126: what's the sampling frequency for the day/night circle?
**Response: The $FCO_2$ measurement frequency was one per hour and $CO_2$ concentration was recorded continuously and averaged for 5 minute interval. Added to revised text.**

L152: change 'location' to 'locations'. – **Fixed.**

Reviewer: Also, it would be helpful to briefly describe the measurement procedures, instead of referring readers to published papers for details. These papers might not be accessible to some of the journal readers.

**Response: We agree and added the following information in the revised text of the section 2.2: $CO_2$ fluxes were measured with two floating chambers equipped with nondispersive infrared $CO_2$ logger (ELG, SenseAir). The $CO_2$ accumulation rate inside each chamber was recorded continuously at 300 s interval. We used first 0.5–1 h of measurements for computing $CO_2$ accumulation rate inside each chamber by linear regression.**

L154: what are the standard approaches? Please clarify and provide details.

**Response: $CO_2$ fluxes were calculated from wind speed and surface water gas concentrations. This technique is based on the two-layer model of Liss and Slater (1974), and widely used for GHG flux assessment (Repo et al. 2007; Juutinen et al. 2009; Laurion et al. 2010; Elder et al. 2018). The gas transfer coefficient was taken from Cole and Caraco (1998):**

$$k_{600} = 2.07 + 0.215 \cdot U_{10}^{1.7} \tag{1}$$

**where $U_{10}$ is the wind speed taken at 10 m height. Average daily wind speed was retrieved from official data of the nearest weather station (Belyi Yar town) as published by Rosgidromet for the day of sampling.**
**We added this missing information, together with details of CH4 measurements in the revised text of section 2.2.**

L156: For flowing streams and rivers, the major driver of the gas transfer velocity is flow velocity, not wind speed.

**Response: This is certainly true for other boreal rivers with high runoff, high flow velocity and pronounced turbulence. The rivers of western Siberian Lowland exhibit slow flow rate, and calculation of the C evasion using river slope (velocity) as performed by Serikova et al. (2018) does not improve the accuracy of $K_T$ calculation because all of the water surfaces of the sampled rivers were considered flat and had a laminar flow. In fact, the water flow was calm and lacked turbulence throughout the river course, even at peak discharge, due to the overall flat terrain of the WSL.**

L181: The DIC concentrations in base flow is even higher than the DOC concentrations (table 1). But here the contribution of carbonate C to total C is only 0.3%. this looks problematic. please double check.

**Response: This is a misunderstanding. The DIC dominated dissolved (< 0.45 µm) load of the rivers during baseflow. However, due to the dominance of peat and clay soils of the river watersheds and lack of carbonate minerals in the river suspended matter (RSM), the concentration of inorganic carbon in the suspended (> 0.45 µm) fraction of the river load was negligibly small compared to that of organic carbon. This observation is fully consistent with previous studies of suspended (Krickov et al., 2019) and dissolved (Pokrovsky et al., 2020) load of other WSL rivers.**

L195: what is the spatial resolution of the biomass and soil OC content datasets?

**Response: The biomass and soil OC content were obtained from BIOMASAR2 dataset in raster format with spatial resolution of 1 x 1 km (Santoro et al., 2010).**

**The soil OC content was taken from the Northern Circumpolar Soil Carbon Database (NCSCD). The original NCSCD dataset produced in GIS vector format corresponding to 1:1000000 scale of topographic map. It could be rasterized to 1 x 1 km pixel resolution [http://www.bbcc.su.se/data/ncscd/ and http://su.diva-portal.org/smash/record.jsf?pid=diva2%3A637770&dswid=1526)]**
**Added to revised text accordingly.**

L219: a lack of systematic change? Note the pCO2 changed by a factor of 2 when tributaries with high CO2 concentrations join the mainstem.
**Response: This is a very good point. We agree that the original sentence was poorly formulated; we intended to state that there was no systematic change in $CO_2$ concentration between the headwaters and the low reaches of the Ket River. The impact of $CO_2$-rich tributaries is indeed clearly seen and we revised the text as necessary.**

L241-247: these are not results, move them to the discussion section.
**Response: We totally agree and shifted this paragraph to the Discussion (section 4.1).**

L297-298: would the precipitation quickly infiltrate into soil and become groundwater?
**Response: Yes, this is certainly possible, notably in the permafrost-free zone of the WSL, as also discussed in L 398-406 of the original manuscript. However the majority of river feeding in the region occurs from bogs/mire at the tributaries, that quickly release the atmospheric water to the hydrological network. This is demonstrated by water isotope study in the WSL (Ala-aho et al., 2018a, b), and discussed in details in section 4.1 (L 373-376).**

L306: as the measurements were performed at the flood peak, this may have caused overestimation.
**Response: We agree that estimations of total C emissions extrapolated to the full period of spring flood should be considered with caution. However we do not expect sizable overestimation of the fluxes; see our detailed response to major comment No 2. It can be seen in Fig. 2 B that May and June exhibit the highest runoff which corresponds to the highest water coverage of the floodplain, as also confirmed by our recent study od the Ob River middle course and its floodplain zone (Krickov et al., 2021).**

L316: how were these %s determined?
**Response: This range reflects both the uncertainty of the water coverage of the territory as analyzed in details by Krickov et al. (2021) based on high temporal and spatial resolution study of C emissions in the floodplain of the river, together with limitations on the seasonal and spatial variations of $CO_2$ emission in the Ket basin assessed in the present study.**

L338-340: why the co2 flux pattern is different from the pco2 pattern?
**Response: Both parameters are directly measured in the field, and strictly speaking, independent of each other. This represents the main added value of the present study compared to previous works where $FCO_2$ was calculated based on hydrochemical measurements in the rivers (Raymond et al., 2013 for example). Enhanced or decreased $CO_2$ evasion measured by floating chambers relative to calculated fluxes can be caused by water turbulence, wind speed and $CO_2$ variations in the air at the river surface. Furthermore, the $CO_2$ concentration measurements encompass quite short period of**

**exposure (typically 5-10 min) compared to fluxes measured by floating chambers; the latter are deployed for 30-60 min period.**

**Note that the difference of the pattern between continuous $pCO_2$ and calculated $CO_2$ flux (green dashed line in Fig. 2A) may stem from the fact that this $FCO_2$ was calculated with $K_T = 4.46$ m d$^{-1}$, from in-situ measured $pCO_2$ values which were averaged over 10-km distance.**

L357-358: Another possible reason is because the measurements were actually not performed in the true headwater streams. All the sites, include the tributary ones, are located along the mainstem and not in the headwater region as shown in Fig. 1.

**Response: We thank the reviewer for pointing out this possibility. We cannot provide a straightforward response to whether the $pCO_2$ increases in the most headwaters compared to the middle course of the tributaries. Note that we typically moved several km upstream of selected tributaries as far as the small boat could go (see Fig. R2 below). Further moving became impossible due to too shallow depths or abundant tree trunks. No need to say that walking in these pristine forest was not feasible. As such we believe that we did our best to tackle still accessible parts of the headwaters, but we acknowledge that further studies are needed to fully address this issue.**

[Figure]

**Fig. R2.** The headwaters of typical small tributaries of the Ket River: Okunevka River (A) and Malaya Anga River (B, C). We could move only several km upstream of tributaries until the tree logs or shallow (30-50 cm) and narrow (1-2 m) channel prevented further progress. Photo credit by Artem Lim.

L366-367: If allochthonous C inputs are the dominant source, pCO2 should have a clear relationship with distance to terrestrial C inputs, i.e., there should be higher pCO2 in tributaries than in the mainstem.

**Response: The reviewer is totally right, and we indeed observed systematically higher $CO_2$ concentration and flux in small tributaries [fed by mire waters with non-processed OM] compared to the main stem; added to revised version. Unfortunately, we could not map in necessary details the tributaries and the main stem watershed to determine the exact distance between the sampling point and potential source of terrestrial C input (specific bog or a floodplain lake). To quantify such a relationship, specially designed study with high spatial resolution (meter to 10 meter pixel size) is needed (such as, for instance, Krickov et al., 2021: https://doi.org/10.1016/j.ecolind.2021.108164) which was beyond the scope of the present work.**

L402: change 'at' to 'in'. – **Fixed.**

L427-452: For these comparisons (similarity and differences), it is quite difficult to follow. Putting them into a table may help. Also, the authors need to make a critical and comprehensive discussion, rather than a general sentence on the possible reasons. This is quite speculative.

**Response: These comparisons are a bit outside of the mainstream of the section (C emission vs export) so that we preferred to strongly shorten this paragraph and remove some irrelevant comparisons. Note that we discuss possible reasons for the observed differences between the Ket River and other boreal rivers in L450-452 of the original version of the manuscript. At the same time, a detailed analysis of environmental physico-chemical, microbiological and landscape factors controlling C pattern in rivers of boreal zone based on other available studies is beyond the scope of the present research (not a review) paper. These literature studies often lack necessary quantitative information on landscape parameters and full hydrochemistry of the water column for each specific watershed. As such, a quantitative comparison with results of the present study is not possible.**

L456: This ignorance may have caused great errors to the annual estimates. Emissions of CO2 during ice melting is exceptionally strong and make a disproportionate contribution to the annual flux estimate

**Response: We agree with this remark; however, in this work we dealt only with open water period. Extensive response to this and second major comment of the reviewer is provided above.**

L460: unclear description of the Ob River.
**Response: Simplified to "The Ob River in the permafrost-free zone"**

L467: change 'thus' to 'this' – **Fixed.**

L502-503: any evidence to support this argument?
**Response: Good point. Here we hypothesized that microbial processing and photodegradation of particulate organic carbon in the water column can be among the main drivers of $CO_2$ supersaturation of the river waters as it is known from field observations and incubation experiments (Attermeyer et al., 2018). These authors demonstrated that riverine POC is 14 times more biodegradable than DOC, and the**

**POC concentration in the Ket River basin increased 4-fold between spring and summer (Table 1, this study). As another support of this argument, we note a local maximum of POC concentration in WSL rivers located at the permafrost thaw boundary (Krickov et al., 2018). This maximum was used to tentatively explain elevated $CO_2$ emissions observed in this part of the WSL, discontinuous to sporadic permafrost zone (Serikova et al., 2018).**

Fig 2: for b&c, change the x-axis to 0-900 for consistency and easy understanding. **- Fixed**

Fig 4e: much higher pco2 during the daytime than the nighttime? Why?
**Response : The reviewer made a good point here, and we thank him/her for pointing this out. After careful analysis of our field work books, we noted that there was quite heavy rainfall, almost full day when the $CO_2$ peak was observed at 7 pm. As such, $CO_2$ mobilization of DOM-rich mire waters from the watershed of the relatively small river Segondenka ($S_{watershed}$ = 472 km$^2$) could explain such a local maximum at the end of the day. Note that the impact of photodegradation of DOM in the water column is unlikely given that end of the day maximum were not observed in other rivers such as Sochur (Fig. 4 A).**

Fig 5d: very low r2, what is the p-value?
**Response: This panel is provided to illustrate a lack of statistically significant (at $p < 0.05$) correlation between $pCO_2$ and wetland coverage of the river watershed, in order to support the statements in L 370-376 of the original text. The p-value here is below 0.5.**

**We thank the Reviewer # 1 for his/her very pertinent remarks and corrections.**

---

## Author Comment (AC2)

**RC2**: ['Comment on egusphere-2022-485'](), Anonymous Referee #2, 30 Sep 2022
General comments

The manuscript entitled "Carbon emissions from Ket River, western Siberia" provides a meaningful contribution to the understanding of carbon export and emissions in the western Siberian Lowland. The title of the manuscript is sufficiently precise and the overall presentation is well structured and clear. Many findings presented in this study are relevant and bring new insights into the processes and controls of carbon processing in this environment. Since the system is influenced by multiple factors, some of the interpretations raised in the discussion are relatively vague or inconclusive. Still, all interpretations and conclusions seem to be well supported by the results. For this reason, I believe that the manuscript will be suitable for publication in "Biogeosciences" after a careful revision.

**We thank the reviewer for positive evaluation of our work and we revised the text following all comments and better argued our interpretations.**

I applaud the initiative of using floating chambers for direct measurements. Although they require more work, the study would provide completely different and much less accurate FCO2 estimations if they weren't employed. Maybe this finding could be emphasized in the abstract or in the final remarks.

**We agree with this proposition and provided relevant information in the Abstract.**

From a cost-effectiveness perspective, I do not see major problems in the approach you took for the final C emissions quantification (especially considering how difficult it is to perform multiple sampling cruises in these areas throughout the year). However, I think that the uncertainty calculation is too simplistic and most probably misleading. I urge the authors to follow best practices recommended at volume 1, chapter 3 of the "2006 IPCC Guidelines for National Greenhouse Gas Inventories" (IPCC, 2006). More specifically, Monte-Carlo approaches (based on probability density functions) have been successfully employed in other assessments. Also, the methods should show all the information required for reproducibility and traceability (e.g. by providing all the equations and later the full data set in online repositories). This does not seem to be the case in this manuscript.

**Following this comment, we added necessary details and equations in section 2.2 as reproduced below:**

**For $CH_4$ analyses, unfiltered water was sampled in 60-mL Serum bottles. For this, the bottles and caps were manually submerged at approx. 30 cm depth from the water surface. The bottles were closed without air bubbles using vinyl stoppers and aluminum caps and immediately poisoned by adding 0.2 mL of saturated $HgCl_2$ via a two-way needle system. In the laboratory, a headspace was created by displacing approximately 40% of water with $N_2$ (99.999%). Two 0.5-mL replicates of the equilibrated headspace were analyzed for their concentrations of $CH_4$, using a Bruker GC-456 gas chromatograph (GC) equipped with flame ionization and thermal conductivity detectors (Serikova et al., 2019; Vorobyev et al., 2021). After every 10 samples, a calibration of the detectors was performed using Air Liquid gas standards (i.e. 145 ppmv). Duplicate injection of the samples showed that results were reproducible within ±5%. The specific gas solubility for $CH_4$ (Yamamoto et al., 1976) were used in calculation of the $CH_4$ content in the water.**

The CO₂ fluxes were measured by using two floating CO₂ chambers equipped with non-dispersive infrared SenseAir® CO₂ loggers (Bastviken et al., 2015), at each of the 7 (spring flood) and 6 (summer baseflow) sampling location of the main stem and 26 tributaries following the procedures described elsewhere (Serikova et al., 2019; Krickov et al., 2021). The chambers were not anchored but slowly free-drifted together with the boat, because it is known that anchored chambers can artificially enhance fluxes due to turbulence thus providing erroneous estimates (Lorke et al., 2015). The CO₂ accumulation rate inside each chamber was recorded continuously at 300 s interval. We used first 0.5–1 h of measurements for computing CO₂ accumulation rate inside each chamber by linear regression. In addition to *in-situ* chamber measurements, CO₂ fluxes were calculated from wind speed and surface water gas concentrations using standard approaches (Guérin et al., 2007; Wanninkhof, 1992; Cole and Caraco, 1998). This technique is based on the two-layer model of Liss and Slater (1974), and widely used for GHG flux assessment (Repo et al. 2007; Laurion et al. 2010; Elder et al. 2018). The gas transfer coefficient was taken from Cole and Caraco (1998):

$$k_{600} = 2.07 + 0.215 \cdot U_{10}^{1.7} \tag{1}$$

where $U_{10}$ is the wind speed taken at 10 m height. Average daily wind speed was retrieved from official data of the nearest weather station (Belyi Yar town) as published by Rosgidromet for the day of sampling. The gas transfer velocity was calculated in two ways - assuming zero wind speed and the actually measured wind speed at the site of sampling or at the Belyi Yar town, middle course of the Ket River.

All the obtained data (full data set) are provided in Supplementary Table S2.

Specifically, we did use Monte-Carlo approach for assessing the CO₂ emissions and dissolved C export fluxes by other Siberian rivers (Serikova et al., 2018; Karlsson et al., 2021; Krickov et al., 2021). We actually did not find any significant (> 20-30%, comparable with inter-annual variability) difference in flux assessment via "classis" simplified approach and the Monte-Carlo (probability density functions). We believe that relative small number of observations (incompatible to recommendations of the IPCC guidelines on much large number of data on GHG inventories) prevent from efficient using of the Monte-Carlo approach in this relatively restricted data set of the Ket River

Estimations for lateral carbon fluxes and POC/DOC are not crucial for most of the conclusions in this paper and seem to be very simplistic and subject to large errors. I recommend authors to reconsider the importance given to the obtained values throughout the text and to improve methods section for a better traceability in this part. **We agree that the lateral riverine export fluxes of DOC and POC do not constitute the central part of this work. However, the fluxes are important for assessing the C emission : export ratio, a fundamental parameter of carbon biogeochemical cycle, especially regarding the response of boreal ecosystems to on-going climate warming. We provided detailed description of the flux calculation in the newly added method section (2.4):**
**The C export flux (May to October) from the Ket basin was calculated based on monthly-averaged discharge at the river mouth in 2019 available from Russian Hydrological Survey and DOC, DIC and POC concentrations measured in the low reaches of the Ket River in this study (see hydrograph in Fig. 1). Annual**

element fluxes should be usually estimated using a LOADEST method (Holmes et al., 2012) from calculated daily element loads. The latter typically obtained from a calibration regression, applied to daily discharge. This calibration regression can be constructed from time series of paired streamflow and measured element concentration data for sufficient period of the year. In our previous works in this and other similar boreal regions, we demonstrated that this method provides reasonable (within 10 to 30 %) agreement with monthly export fluxes calculated by multiplying mean monthly discharge by mean monthly concentration (Chupakov et al., 2020; Pokrovsky et al., 2022; Vorobyev et al., 2019), at least for the WSL territory. Given that the intrinsic uncertainties on mean monthly discharge are also between 20 and 30 % (see discussion in Pokrovsky et al., 2020), in this study, for open-water period export flux calculation, we used DOC, DIC and POC concentrations measured during spring flood (for May and June period) and baseflow (for August, September and October period). For the month of July, we used the mean concentrations of end of May and August-September which is in accord with seasonal discharge pattern of the Ket River. Note that the contribution of non-studied October month to total open water period water flux is < 10 % and thus cannot provide sizable uncertainties.

Our main argument on the validity of the validity of approach employed for lateral export flux assessment is that the lateral C loss (yield) for the Ket River (3.7 t C $km^{-2}_{land}$ $y^{-1}$) is in agreement with regional C (DOC+DIC) yield by permafrost-free small and medium size rivers of the WSL (3 to 4 t C $km^{-2}_{land}$ $y^{-1}$, Pokrovsky et al., 2020) and with the Ob River in the permafrost-free zone (3.6 t C $km^{-2}_{land}$ $y^{-1}$, Vorobyev et al., 2019). Note that the latter study of the Ob River, which is very similar in the environmental context to the Ket River, included high frequency weekly sampling over several years of monitoring. Thus, the similarity of lateral export fluxes of the Ket and Ob Rivers support the validity of approaches for sampling and C yield calculation employed in the present study.

Responses to specific comments:
Lines 34-35 = Poorly known?
**Yes, corrected accordingly.**

Lines 42-43 = Please consider also including the pCH4 ranges.
**0.05 to 2.0 μmol $L^{-1}$; added accordingly**

Lines 50-54 = Please consider revisiting these last sentences after a careful revision of the methods employed in the uncertainty calculations. I think it is important to be very clear on what are the limitations of these estimations right in the abstract to avoid poor usage of the emission values. For example, you mention in lines 50-51 that "C emission from the Ker River basin was estimated to 127+-11 Gg C y-1", however, you've discarded important hot moments/spots, soil emissions/uptake, etc. I guess you should use another term instead of "River basin" here.
**This is very pertinent remark. By this sentence we intended to say "C emissions from the fluvial network (main stem and tributaries) of the Ket River". We also added a sentence on the uncertainties of our conservative estimations and indicated a need for better spatial and temporal resolution**

Lines 73-83 = please consider including some of the values instead of presenting this information in a more qualitative way.

**Abrupt permafrost thaw may release 80 Pg C and gradual thaw may release up to 200 Pg C by 2300 (Turetsky et al., 2020); added accordingly.**

Line 113 = I am not sure if "-0.6..-0.9°C" is a proper way of presenting the temperature range.

**We checked the climate data and stated that the MAAT is -0.7 ±0.1 °C.**

Line 201 = I am not a native English speaker, but "wetted streams" doesn't seem right.

**Here we intended to state "wetted" in contrast to "active"; more precise term is "temporary non-active streams"**

Line 226 = Please consider including the pCH4 ranges.

**The range of $CH_4$ concentration is from 0.05 to 2.0 µmol $L^{-1}$**

Line 244 = This may be a bit far-fetched, but what about emissions linked to vegetation or other hot spots that helps gas leakages? I know this is a completely different context, but something like seen in floodplain trees (e.g. Pangala et al., 2017), maybe? Also, some pictures of the river and streams in the supplementary material would help readers to have a better idea of the environment.

**This is very pertinent remark. This part of the text was moved to the Discussion and we added the possibility of emissions from flooded trees, not investigated in this study.**

**In the revised Fig. S4, we also presented typical environments of the main stem and tributaries during the spring flood: flooded birch forest (A), abundant grassland (B), and tree-sheltered main stem (C) as illustrated below:**

[Figure]

**Fig. R1. Typical landscapes of the Ket River and tributaries during spring flood (May 2019)**

Lines 376-380 = To me it seems that you have raised a hypothesis (fluxes comes from bog water), tested it (calculate the bog area) and the results "falsified" your hypothesis. Shouldn't you then present an alternative hypothesis here?

**This is true. We could not find any other single 'alternative' factor capable of describing the emission pattern. We added the following text in the revised Discussion:**

**The main unexpected result of this study is that none of the physiochemical parameters of the water column and the landcover factor is sufficiently strong to drive the $CO_2$ and $CH_4$ patterns, although they show pronounced spatial and seasonal variations. A likely explanation is that simultaneous operation of multiple aquatic processes that include carbon, oxygen, nutrient, and plankton and peryphyton dynamics as well as sediment respiration control the $CO_2$ and $CH_4$ exchanges with the atmosphere, as it is known for boreal lakes and floodplain zones of the boreal rivers (i.e., Bayer et al., 2019; Zabelina et al., 2021; Krickov et al., 2019). Given that even a multiparametric statistics (PCA) did no demonstrate sizable explanation capacity of the data set, we cannot exclude that these potential physico-chemical, microbiological and landscape drivers are working in different (opposing) directions and have counteracted each other.**

Lines 381-386 = Does it has any relationship with increased primary productivity per area inland? Any estimates?

**Good point. Unfortunately, we do not have primary productivity data with sufficient spatial resolution to test this hypothesis, because the size of the Ket River tributaries is quite small. Currently, this work is in progress at the scale of much large Siberian watersheds. Note that the total vegetation biomass of the catchment positively (but not significantly at $p < 0.05$) correlated with $CO_2$ concentration and fluxes in other western Siberian watersheds (Krickov et al., 2022, submitted to Sci. Total Environ).**

Line 456 = Also mentioned "Ket basin", I guess this is inaccurate.

**Thank you pointing this out. Yes, here we meant fluvial network of the Ket River; corrected accordingly.**

**We thank Reviewer # 2 for his/her very pertinent and useful comments.**

---

## Author Comment (AC3)

1) GENERAL COMMENTS

Lim et al. report a high-quality data-set of CO2 and CH4 concentration measurements in the Ket River in Siberia obtained during high-water and low-water. This is a very useful contribution to on-going efforts to collect data to better evaluate the carbon emissions from inland waters because the studied river drains a remote and nearly undisturbed (pristine) watershed dominated by peat bog and taiga forest. Unfortunately, the analysis is (in my opinion) not well structured and the authors might want to spend some extra time on thinking through how to present and analyze the data, and profoundly re-structure the paper and streamline the present content.

For instance, the authors computed the fluxes of CO2 with a gas transfer velocity parameterization for lakes; this gave (unsurprisingly) very different results from the fluxes of CO2 measured with floating chambers. This was predictable and in my opinion not very useful, just distracting.

**There is a misunderstanding concerning the origin of $K_T = 4.46$ m d$^{-1}$. This value was used for consistency with large Siberian rivers (Karlsson et al., 2021; Vorobyev et al., 2021), in agreement with world average for rivers of low velocity (Raymond et al., 2013). However, the fluxes obtained by the wind speed method are in more reasonable agreement with chamber measured fluxes (Table S2 of the Supplement): the calculated FCO$_2$ are generally 1.5 to 2 times higher than the measured values, but in 30% of cases the wind-speed calculated fluxes are similar to or lower than those measured by floating chambers.**

Regarding formal aspects, the authors should spend some extra time producing high quality figures. Figure 2 is extremely confusing and does a very poor job at presenting this data-set that required a lot of effort to acquire. Figure 3 shows some nice patterns of pCO2 and CH4 concentration in terms of seasonal variations (high-water vs low-water) as well as in terms of stream size (main-stem vs tributaries). A more straightforward and attractive presentation and discussion could be built on these simple patterns. Instead, this nice and potentially interesting information is diluted in a lot of rather unnecessary elements such as computations of fluxes with inadequate gas transfer parameterizations and correlations with not very useful variables such as total bacterial counts (see comments below).

**We removed all unnecessary information while adding the recommended citations and stream focusing the discussion of the results.**

2) MAIN COMMENTS

L37 and L218: I'm unsure that the term "continuous" applies to measurements of CO2 to this study. My perception of "continuous measurements" is that water is continuously pumped through an equilibrator system connected to a CO2 detector (or equivalent setup) and then the data are logged at regular intervals (1 min or less) (Abril et al. 2014; Crawford et al. 2016b; 2017 Borges et al. 2019). This means that the measurement of CO2 is not interrupted for long periods (and runs for a few hours to a few days) while the boat is sailing. The authors made discrete samples with the boat stopped at a given spot. Albeit they made numerous measurements this should qualify as discrete sampling and not continuous. This is not just a semantic issue; the authors made 764 pCO2 measurements over the distance of the boat route (834 km) as stated L 218. This roughly corresponds to one measurement every 1 km. This is

still quite coarse to describe extremely dynamic river systems. As an example, Borges et al. (2019) showed very marked cross-channel gradients of CO2 in the mainstem Congo River, corresponding to a spatial scale of the order of 1 km (using what truly qualifies as "continuous").

**We totally agree that the floating chamber measurements performed in the present study are discrete and not continuous. However, during the spring flood period, the $CO_2$ concentration was, indeed, measured continuously. As it is stated in the text, a Campbell logger was connected to the submerged $CO_2$ sensor system allowing continuous recording of the $CO_2$ concentration, water temperature and pressure every minute. We specified that these readings were averaged over 10 minute intervals yielding 732 individual $pCO_2$, water temperature and pressure values. At the same time, we admit that identifying and quantifying local-scale hot spots and hot moments of $CO_2$ release or uptake were not within the objectives of our study.**

L150: The authors measured CO2 fluxes between water and air with floating chambers. Lorke et al. (2015) have shown that anchored chambers enhance turbulence under the chambers and artificially enhance fluxes, thus providing erroneous estimates. Please specify if the chambers used in the present study were anchored or free-drifting. If the chambers were anchored then the data should used with extreme caution, especially for the flood period when presumably the flow was higher. In my opinion, these chamber measurements are not necessary, and fluxes should be computed from gas transfer velocity using an adequate parametrizations applied to spatial data, please refer to Liu et al. (2022).

**The reviewer made a good point here. The chambers were not anchored but allowed to move together with the boat. We believe that chamber measurements are most valuable contribution of the present study, also noted by other reviewers. In the Supplementary Table S2, for interested reader, we provided results of flux calculations by different methods, assuming zero wind speed, actual wind speed and average $K_T$ of 4.46 m d$^{-1}$ for the WSL rivers. The latter value is reported merely for consistency with previous global estimations of $CO_2$ emissions from great Siberian rivers and their tributaries such as Ob (Karlsson et al., 2021) and Lena (Vorobyev et al., 2021). At the same time, all the interpretations and correlations presented in our discussion are based on actually measured chamber-based fluxes.**

L154: The authors also computed the CO2 fluxes between water and air from CO2 concentrations and the gas transfer velocity. The cited references (Guérin, et al., 2007; Wanninkhof, 1992; Cole and Caraco, 1998) provide parameterizations for lakes that are inadequate for computing the gas transfer velocity in running waters. The authors provides these 3 references, although it was unclear to me which one was actually used in the computations. The gas transfer velocity in streams and rivers can be derived from stream flow and stream slope, that in turn can be derived from spatial data; please refer to Liu et al. (2022).

**We agree that lake parameters are not always suitable for computing the gas transfer velocity in running waters. However, the Ket River and especially its tributaries exhibit quite low slope and velocity and the waters are often stagnant due to extremely flat terrain. For convenience, we calculated the fluxes for different wind speed and these fluxes were in reasonable agreement with those measured by floating chambers. Moreover, the range of $K_T$ obtained in this study for the Ket River basin is consistent with that reported based on multiple measurements and calculations using stream flow and stream slope approach (1.2 – 1.5 m d$^{-1}$) by Serikova et al. (2018). Although the latter work did not encompass the Ket River main steam and tributaries, the permafrost-free**

**WSL rivers studied by Serikova et al. (2018) are highly similar to those of the Ket catchment.**
**Noteworthy that the $k_T$ values calculated for western Siberia by Liu et al. (2022) based on reach-level slope and flow velocity (i.e., below or equaled to 2 m d$^{-1}$) are just in excellent agreement with those obtained in the present study with chamber measurements (Table 1 and Table S2). A likely explanation is relative low values of gas transfer velocity measured in the small streams of the Ket basin in this study (0.2 - 2.0 m d$^{-1}$, Table 1). These values are typical of lakes rather than rivers (i.e., Kokic et al., 2015) and stem from low flow rate, strongly forested and wind-protected river bed without distinct valley due to extremely flat orographic context of this part of the WSL (Serikova et al., 2018).**

L 216: The authors state that there are no spatial variations in CO2. I suggest to mention here that CO2 in tributaries was higher than in the main stem. This corresponds to a "systematic" pattern of variation.
**We totally agree and added this important information in the text. Note that, while $CO_2$ concentrations were different between tributaries and the main stem during both flood and baseflow, the $CO_2$ flux was not different between the main stem and tributaries regardless of season, as was assessed by Mann-Whitney U test (Table S4 B).**

Also, I suggest that the authors extract the Strahler order of the sampled streams and rivers and analyze if there are differences by stream size. It is quite frequent that lower order streams show higher $CO_2$ values and higher order (Butman and Raymond 2011; Borges et al. 2019), although not always necessarily the case (Borges et al. 2018). Stream size could also be analyzed in terms of catchment area, in addition to Strahler order. Stream size can be used also for upscaling concentrations and fluxes, refer for example to Borges et al. (2019).
**This is very valuable comment. We did examine the impact of stream size (catchment area) on $CO_2$ concentration and fluxes and found that $CO_2$ concentrations (but not fluxes) increased with a decrease of the river watershed area (Table 2). As such, it was not necessary to take into account the stream order for upscaling the C emissions from the Ket River basin.**

**RESPONSES TO SPECIFIC COMMENTS of Reviewer No 3**

L 34 : I suggest to define « medium–size rivers »
**50,000 to 300,000 km², added accordingly**

L 34 : I suggest to remove « poorly » or replace by « largely » but « poorly unknown » is ackward.
**Here we intended to say "poorly known"; corrected**

L 40: I suggest to mention the months-years of sampling
**May 2019 and end of August - beginning of September 2019**

L40: I suggest to replace "CO2 concentration" by partial pressure of CO2.
**Agree and corrected accordingly**

L40-41: I suggest to mention the differences in pCO2 between base flow and flood period.
**In the tributaries, the $pCO_2$ was 40% higher during baseflow compared to spring flood, whereas in the main stem, it did not vary significantly across the seasons**

L41-43: I suggest to provide the range of the CH4 concentrations values rather than the ratio to CO2.

**The CH$_4$ concentrations ranged from 0.05 to 2.0 µmol L$^{-1}$; added accordingly.**

L 47 : I suggest to specify if this is this spatial or temporal "variability" ? or both ?

**Both spatial and seasonal variability; added to the revised text**

L 49 : The hypothesis of lower path soil-water CO2 inputs during summer is based on what ? During summer-time numerous processes contribute to increase CO2 in rivers compared winter such as higher temperature stimulating microbial metabolism, longer residence time and lower gas transfer velocity (lower river flow), in addition to changes in flow paths of soil-water flows (Borges et al. 2018).

**The underground waters produced by dissolution of carbonate mineral-bearing rocks of the Ket catchment are better connected to the river during summer baseflow compared to spring flood. This is well established from former works on the WSL hydrochemistry across seasons (Pokrovsky et al., 2015, 2020). However, we totally agree with the reviewer that multiple processes acting in parallel can contribute to increased CO$_2$ in rivers in summer compared to early spring. To avoid speculations, we removed this sentence from the Abstract but discussed the works of Borges et al. (2018) in the revised section 4.1.**

L51: "lateral" usually refers to exchange between river and riparian zones (e.g. floodplains). Term "downstream C export" might be more adequate. I suggest to specify if this downstream C export refers to inorganic, organic or total carbon and if dissolved or dissolved+particulate.

**This is very pertinent remark. We consider the total downstream C export which includes DIC, DOC and POC. We stated this in the discussion section and revised the Abstract accordingly.**

L67: define abbreviation pCO2

**Partial CO$_2$ pressure; added to the text.**

L69: This statement does not reflect current state of CO2 studies in rivers. There is a fast growing very large amount of studies reporting directly measured CO2 measurements either discretely (Alin et al. 2011; Borges et al. 2015; Amaral et al. 2018; 2022; Leng et al. 2022), continuously at fixed sites (Crawford et al. 2016a, Schneider et al. 2020; Gómez-Gener et al. 2021), and continuously underway (Abril et al. 2014; Crawford et al. 2016b; 2017; Borges et al. 2019). And this is also the case for studies in "under-represented or ignored regions" as stated, and for more than a decade (Alin et al. 2011).

**We certainly agree with this remark. Moreover, the most recent study of Liu et al (2022) deals with directly measured CO$_2$ at the world wide scale. However, to the best of our knowledge, there is no information on measured CO$_2$ concentration in Siberian rivers other than of our group on the Lena and Ob Rivers (Vorobyev et al., 2021; Karlsson et al., 2021), limited data on the Kolyma River (Denfeld et al., 2013), and small WSL rivers across a permafrost gradient (Serikova et al., 2018). Therefore, by under-represented regions we meant all northern Eurasian territories between Scandinavia and Alaska. We modified the text for clarity and cited the useful papers noted by reviewer.**

L 71-72: This is correct and there are some studies available (Abril et al. 2014; Crawford et al. 2016b; 2017 Borges et al. 2019). It could be useful to briefly mention if there is and what is

the added value to make continuous "regional high spatial resolution measurements" of CO2 compared to discrete measurements, based on past published papers.
**We totally agree and cited the relevant papers in the revised text of the Introduction. We would like to underline that all available studies are limited to tropical and temperate zones of the world, and boreal regions of Western Europe and Northern America, and thus, further continuous and discrete measurements of $CO_2$ concentration and fluxes of under-represented regions such as Northern Eurasia are needed.**

L73-74: Please clarify what do you mean by "High latitude regions are important". With respect to total CO2 emissions at global scale, rivers in high latitude regions are not important according to the study of Liu et al. (2022) who show that "tropical rivers are responsible for 57% of the global emission, more than temperate and Arctic regions combined (30 and 13%, respectively)".
**Good point. Our argument here is not the current $CO_2$ emissions from high latitude regions, but future climate warming scenario, according to which boreal and permafrost-affected regions can release significant amount of soil C which is likely to be transformed into GHG in the aquatic systems.**

L113: there's some sort of typo here " 0.6..-0.9 °C"
**Thanks for pointing this out. The MAAT is  -0.7 ± 0.1 °C.**

L 148 : For a journal such as Biogeosciences I think it is insufficient to refer to other papers for basic methodological information. I suggest to provide details on the gas used for the headspace, on the calibration gases, on the detection limit, precision and accuracy. It could also be useful to mention the typical time interval between sampling and analysis.
**This is important remark, also raised by other reviewers. For $CH_4$ analyses, unfiltered water was sampled in 60-mL Serum bottles. For this, the bottles and caps were submerged at approx. 30 cm depth from the water surface. The bottles closed without air bubbles using vinyl stoppers and aluminum caps and immediately poisoned by adding 0.2 mL of saturated $HgCl_2$ via a two-way needle system. The samples were stored approximately one week in the refrigerator before the analyses.  In the laboratory, a headspace was created by displacing approximately 40% of water with $N_2$ (99.999%). Two 0.5-mL replicates of the equilibrated headspace were analyzed for their concentrations of $CH_4$, using a Bruker GC-456 gas chromatograph (GC) equipped with flame ionization and thermal conductivity detectors (Serikova et al., 2019; Vorobyev et al., 2021). After every 10 samples, a calibration of the detectors was performed using Air Liquid gas standards (i.e. 145 ppmv). Duplicate injection of the samples showed that results were reproducible within ±5%. The specific gas solubility for $CH_4$ (Yamamoto et al., 1976) were used in calculation of the $CH_4$ content in the water.**

L129-139: Similarly for $CO_2$ please provide information on precision. Is the stated accuracy given by the manufacturer or was this determined by the authors? Also specify how the Vaisala instrument was calibrated. Did you trust the factory calibration or did you carry out calibration in the lab? Was the probe checked for signal drift before and after the cruise against standards ? Did you measure atmospheric $CO_2$ with the Vaisala probe during the cruises as a check of good functioning ?
**We agree that referring to previous publications for detailed description of analytical techniques is not appropriate. The calibration of the sensor was our priority during this study. Sensor preparation was conducted in the lab following the method described by Johnson et al. (2009). The measurement unit (MI70, Vaisala®; accuracy ±**

**0.2%) was connected to the sensor allowing instantaneous readings of $p$CO$_2$. The sensors were calibrated in the lab against standard gas mixtures (0, 800, 3 000, 8 000 ppm; linear regression with R$^2$ > 0.99) before and after the field campaign. The sensors' drift was 0.03-0.06% per day and overall error was 4-8% (relative standard deviation, RSD). Following calibration, post-measurement correction of the sensor output induced by changes in water temperature and barometric pressure was done by applying empirically derived coefficients following Johnson et al. (2009). These corrections never exceeded 5% of the measured values. During the cruise, we routinely measured atmospheric CO$_2$ with the probe ac a check for its good functioning. Furthermore, we tested two different sensors in several sites of the river transect: a main probe used for continuous measurements and another probe used as a control and never employed for continuous measurements. We did not find any sizable (>10%) difference in measured CO$_2$ concentration between these two probes.**

L 144 : how was the water sampled and transferred to the serum vials ? With some sort of sampling bottle ? Niskin or equivalent ?

**For CH$_4$ analyses, unfiltered water was sampled in 60-mL Serum bottles. For this, the bottles and caps were manually submerged at approx. 30 cm depth from the water surface. The bottles were closed without air bubbles using vinyl stoppers and aluminum.**

L165: I suggest to define the "NIST" abbreviation

**NIST is for National Institute of Standards and Technology.**

L189-193: Please specify if the land cover data correspond to the whole catchment area upstream of the sampling point or if this corresponds to the riparian vegetation just adjacent to the sampling point.

**The land cover data correspond to the whole catchment area upstream of the sampling point.**

L 216 : I suggest to remove word « emission ». You cannot pre-suppose an emission, some rivers on some occasions can be sinks of CO$_2$ (Crawford et al. 2016b).

**This is totally, true; we agree with this remark.**

L 246: I'm not sure this "warning" is useful since the authors used a parameterization for lakes, and this was not a very good idea to start with.

**We agree that this text is not at right place; this belongs to the Discussion. Results of the present study clearly demonstrate that high value of K$_T$ (transfer velocity), pertinent to large Siberian rivers of the permafrost regions, can not be used for the Ket River and tributaries located in the boreal permafrost-free zone, due to slow flow rate and strong shading of the river bed by dense taiga forest, leading to quite short fetch. This has to be taken into account for Pan-Siberian upscale of emission fluxes. We reorganized this part in the revised version.**

L 295 : It's quite unusual to look into the effect of catchment lithology on fluvial CO$_2$ and CH$_4$ concentrations. Lithology will affect the HCO3- content and DIC content, but with little direct impact on CO$_2$ levels and certainly not on CH$_4$. I suggest the authors restrict this analysis to DIC (or remove altogether this analysis that is just a distraction).

**We totally agree that one cannot expect direct lithological control on CH$_4$ concentration in the river water (other than via pH buffering by carbonate rocks of the catchment). However, the carbonate rocks/concretions present in the mother rock can strongly affect**

the CO₂ pattern, via notably underground discharge of DIC-rich waters during baseflow. This is consistent with absence of correlations during high flow (spring flood), when the rocks are essentially disconnected from the river. We would like to recall that first assessments of fluvial CO₂ emissions (Raymond et al., 2013) were largely based on pCO₂ values calculated from DIC+pH of the river waters. The latter parameters are directly controlled by the proportion of carbonate rocks on the catchment…
**Following the recommendation of the reviewer, we greatly diminished the presentation of lithological aspects in the revised version.**

L297-298: This is also quite unusual. I would envisage seasonal variations precipitation to explain seasonal variations of CO2, but not spatial variations during a given period, in this case base flow. Correlation does not necessary imply causation, some correlations are spurious or indirect. There's a possibility that this is relate to stream size, as precipitation at catchment scale, also captures catchment surface area in an area of relatively homogeneous precipitation. I suggest to remove altogether this analysis that is just a distraction.
**We agree that correlation does not necessary imply causation, and we alerted the reader in the revised version. Note that we also attempted Principal Component Analysis (PCA) which, however, did not allow any better identification of possible driving factors. Therefore, we cannot exclude that these potential physico-chemical, microbiological and landscape drivers are working in different (opposing) directions and have counteracted each other.**
**Our tentative explanation for positive correlations between mean annual precipitation (MAP) and pCO₂ and FCO₂ during the baseflow is that they could reflect the importance of water storage in the mires and wetlands during the summer time, and progressive release of CO₂ and DOC-rich waters from the wetlands to the streams. Note that there is a positive relationship between pCO₂ and river catchment area, which is in line with proposition of the reviewer.**

L 346: The paper of Gómez-Gener et al. (2021) gives a reasonably good account of diel variations of pCO₂ in temperate rivers but reports measurement in an extremely limited number of sites in tropical rivers. So this study does not allow to make generalizations on "tropical rivers". There are other studies in tropical rivers that have shown that diel variations of CO₂ are undetectable such as the Congo (Borges et al. 2019) because aquatic pelagic primary production is low (Descy et al. 2018) due to strong light attenuation the water column by DOM.
**This is an excellent comment, which helps a lot to explain the lack of variation in the case of the Ket River which also has high aromatic DOC content. Note that many streams considered in Gómez-Gener et al. (2021) work are low in DOC, or this DOC is essentially autochthonous. We thank the reviewer for pointing out these important papers on tropical rivers and we carefully revised the text and better argued our explanations.**

L363-367: This is a reasonable explanation. However, "homogeneous landscape" and "strong allochthonous sources of organic carbon" can still lead to variations of CO₂ per stream size, with small systems showing higher values than large systems as predicted conceptually (Hotchkiss et al. 2015) and verified at basin-scale (e.g. Borges et al. 2019).
**We totally agree with this remark. Indeed, the SUVA and bacterial number (TBC) positively correlated with both pCO₂ and FCO₂ during summer (Fig. 5 A, B), which may indicate non-negligible role of bacterial processing of allochthonous (aromatic) DOC delivered to the water column from wetlands and mires. Consistent with this, we**

**observed systematically higher CO₂ concentration and flux in small tributaries [which were fed by mire waters with 'non-processed' OM] compared to the main stem (Table 2).**

L 381: I suggest to remove the word "interesting". This is self-evaluation, let the readers decide what's interesting. Same applies to word "notable" L 361.
**We agree with these remarks.**

L 477-515: Section "Concluding remarks" provides a summary of the paper and thus duplicates the content of abstract. This section could be removed or streamlined.
**We agree and greatly revised this section via shortening it and focusing on most important findings and perspectives.**

In Figure 2, I suggest to show the « continuous » pCO₂ measurements data points as a discrete symbols (dots) rather than a line.
**With spatial resolution of this figure, dots representing continuous pCO2 reading would be shown as a line. We improved the spatial resolution of this figure following the recommendation of the reviewer.**

Figure 2 is incredibly confusing and in my opinion undermines the large sampling effort. I suggest to make separate figures for pCO2 and FCO2 and not try to show all of the data together in single plot. Please provide a graphical representation of the pCO2 during the flood period. If I understand correctly the symbols, the blue diamonds in plot A) are for the FCO2 and not pCO2 in the tributaries. But Table 1 shows that pCO2 was measured in the tributaries during the flood period. I also suggest to remove the "continuous FCO2". The term is misleading since it's FCO2 computed from "continuous" pCO2. Also since the figure mixes FCO2 measured with the chambers and computed with a gas transfer velocity and that the values are very different, the impression given by the figure is very confusing.
**We thank the reviewer for noted inconsistences in this figure: these were now corrected. We basically agree with this remark; we removed the fluxes computed with fixed gas transfer velocity. We believe that calculated 'continuous' fluxes are useful. In the revised version, these fluxes were calculated based on directly measured pCO₂ and K$_T$ values calculated as an average of two adjacent chambers, instead of fixed or wind-based K$_T$ value.
We strongly revised this figure as following:**

[Figure]

**Figure R1.** The measured pCO$_2$ (**A, C**) and CO$_2$ fluxes (**B, D**) during spring flood (**A, B**) and summer baseflow (**C, D**) of the Ket River main stem and tributaries (over the 830 km distance, from the headwaters to the mouth (left to right). The symbols represent discrete in situ pCO$_2$ (Vaissala) and FCO$_2$ (floating chambers) measurements of the main stem (red circles) and tributaries (blue diamonds). Continuous in-situ pCO$_2$ measurements and calculated FCO$_2$ are available only for the main stem in spring (black crosses). For the latter, we used an average value of gas transfer velocity (k$_T$) between two chamber sites (separated by a distance of 50 to 100 km) to calculate the FCO$_2$ from in-situ measured pCO$_2$ in the river section between these two sites. Note that during summer baseflow, the water level did not allow reaching the headwaters of the Ket River (first 0-200 km on the river course).

Figure 5 : pCO2 should be in the Y-axis and the potential predictors/descriptors (SUVA, land cover) in the X-axis.
The correlation of pCO2 and TBC in Fig. 5B is weak and not very informative. The TBC only informs on the presence of microbes and not their activity. Also, if CO2 comes from soil-water as suggested by the authors then it is not produced in-stream and we should not expect a correlation with TBC. This cannot go both ways.
**We agree with these remarks and strongly reorganized this figure as following, via presenting only most significant (p < 0.05) correlations:**

[Figure]

**Figure R2.** Significant ($p < 0.05$) control of dissolved oxygen (A), $SUVA_{254}$ (B), light needleleaf forest (C), and mean annual precipitation (D) on CO2 concentration in the Ket River and tributaries during summer baseflow.

**We thank Reviewer # 3 for his/her very pertinent and useful comments.**

---

## Author Response (AR2)

**Dear Dr Gwenaël Abril**

**The second revision of our manuscript received very constructive critics, and we carefully addressed each comment of the reviewer in the newly revised version.**

You stated that "Wind speed parametrization of gas transfer velocities for lakes cannot be applied in a river, even with low slope and low current velocities and without any justification." **We totally agree with this comment; in our work we did not use for presentation of Results and Discussion the wind speed calculated gas transfer velocities, neither the fixed $K_T$ used previously for large Siberian rivers. Results of alternative methods of flux calculation were presented only in the Supplement, and this was done for consistency with other works in this field (including the most complete one of Castro-Morales et al. (2022)). Now we removed all mentioning of wind speed parametrization and relevant numbers from the revised manuscript and the Supplement.**

**Following your recommendation, we calculated gas transfer coefficients for sampled rivers taking into account hydraulic variables and compared them with those obtained in the field with drifting chambers. The relevant discussion has been added to L 423-443 of the section 4.1. Again, we would like to underline that all the interpretation of the data were based on chamber-measured fluxes.**

**Responses to Reviewer No 3**

The authors have made some improvements on the paper, but I think they could still make an extra effort to bring this work to the standards of Biogeoscience and that the paper does justice to the enormous effort in acquiring these data.

**We further revised our manuscript and performed necessary calculations of transfer coefficients.**

I still think it is useless and unjustified to use the parameterization of Cole and Caraco (1998) developed for lakes to compute fluxes in a river. This just adds to confusion and distraction to the paper. On the other hand it would be extremely useful to compute the gas transfer velocity from slope and flow using parameterization for rivers as used by Liu et al. (2022). This approach is applicable to all rivers (large and small) even those with low flow and low slope as Ket river. Liu et al. (2022) explain in detail how to make the computations, this can be achieved without too much effort with some GIS modelling. The authors could then use the gas transfer velocity from the chamber measurements to validate the modelled values.

**We agree with this remark and removed all the parametrization of Cole and Caraco (1998), developed for lakes, from the revised manuscript. The only reason of presenting Cole and Caraco (1998) method was for the consistency with other works on Siberian rivers, in particular, the Ambolikha River (Castro-Morales et al., 2022). In the newly revised version, we performed the $K_T$ calculations following the work of Liu et al. (2022) as requested by the reviewer as described in L 433-441 of the revised Discussion. Specifically, gas transfer velocity (k) was estimated from channel slope (S) and flow velocity (V) using either equation (4) of Raymond et al. (2012)**

$$k_{600} = (VS)^{0.76} * 951.5$$

**or equation (5) of Raymond et al. (2012):**

$$k_{600} = 2841 \, SV + 2.02$$

**Both equations have been shown to predict reasonable k over large spatial scales in various regions as reviewed by Liu et al. (2022).**

Raymond, P. A., Zappa, C.J., Butman, D., Bott, T.L., Potter, J., Mulholland, P., Laursen, A. E., McDowell, W. H., and Newbold, D.: Scaling the gas transfer velocity and hydraulic geometry in streams and small rivers. *Limnology and Oceanography: Fluids and Environments*, *2*(1), 41-53. https://doi.org/10.1215/21573689-1597669, 2012.

**The table R1 below lists the obtained values, in comparison to chamber-measured transfer coefficients, as median and IQR for all rivers of the Ket basin (main stem and tributaries):**

**Table R1.** Chamber-measured and calculated transfer coefficients for all studied rivers of the Ket basin.

| $K_T$, m d$^{-1}$ | median | IQR | $Q_1$ | $Q_3$ |
|---|---|---|---|---|
| Measured by chambers | 0.69 | 0.76 | 0.47 | 1.23 |
| Raymond et al., 2012, eq. 4 | 1.02 | 1.25 | 0.27 | 1.52 |
| Raymond et al., 2012, eq. 5 | 1.81 | 0.51 | 1.69 | 2.19 |

**For convenience, we illustrated results of these comparison in Fig. R1 below:**

[Figure]

■ Measured
■ Raymond et al., 2012, eq. 4
▨ Raymond et al., 2012, eq. 5

**Fig. R1. Comparison of gas transfer coefficients for all rivers and streams of the Ket basin, measured by floating chamber and calculated using average channel slope and flow velocity, using two different equations of Raymond et al. (2012). Note a very good agreement between measured and calculated (Eqn. 4 of Raymond et al., 2012) transfer coefficients for the rivers of the Ket basin.**

The use of multiple gas transfer velocity values is confusing, and this is worsened by the fact that the legend of figures and Table that are not explicit. Table 1 and Figure 3 report FCO2 and FCH4 and because the legend is very short it is not possible to know just by reading the table/figure what these values correspond to. Are these the chamber measurements (for FCO2) or the computed values ? If these are the computed values how were the computations made ? With the constant Kt of 4.46 m d-1 "representative for large lowland rivers" (L207), the gas transfer velocity computed from wind with Cole and Caraco, or using an average K derived from chambers ?
**We are sorry for not being sufficiently explicit about data presentation. Throughout the main text, tables and figures, we always used only chamber-measured fluxes and measured gas transfer coefficients. Figure 3 and Table 2 present chamber-measured CO$_2$ fluxes; we added this information in the figure legend and table caption.**

There is (in my opinion) very little gain in computing fluxes with all these different methods. On the contrary this is just a distraction and source of confusion for the readers.
**We totally agree and removed all mentioning of different methods of flux calculation. In the newly revised version, we present only chamber-measured CO₂ fluxes.**

I still think it would be useful to extract Strahler order and plot the data as function of Strahler order rather than bundling all of the data into "tributaries". This is quite useful approach, check figure 2 of Butman & Raymond (2011, https://www.nature.com/articles/ngeo1294). Given the enormous amount of work to acquire the data, the authors might want to spend a couple of hours extracting Strahler order that can be computed with DEM data and GIS tools such as Quantum GIS (freeware).
**The Strahler order of studied rivers ranges from 2 to 9; added to the text (L 113-114) and revised Table S1. Following the reviewer's suggestion, we plotted the pCO₂ and chamber-measured FCO₂ and K_T as a function of stream order (Fig. S5 of revised manuscript), presented for convenience in Fig. R2 below:**

[Figure]

**Fig. R2**. Distribution of $CO_2$ concentration (**A**), chamber-measured flux (**B**) and chamber-based gas transfer velocity (**C**) across stream orders in the Ket River basin during spring flood (green squares) and summer baseflow (red circles).

**As expected at such low slopes and extremely flat terrain, there is no impact of stream order on CO₂ emissions regardless of season.**

I still think it is superfluous to present in table 2 the correlations of fluvial CO2 and CH4 with lithology. There is no established link between lithology and CH4. Lithology affects HCO3- content but not CO2. The fluvial CO2 content depends mainly on respiration not lithology. The respiration that leads to CO2 in streams occurs in soils and/or in-stream. This is why Lauerwald et al. (2015, doi:10.1002/ 2014GB004941) modelled CO2 in rivers using catchment net primary production and not lithology. This is why CO2 in rivers correlates with DOM rather than Ca2+.

**We agree and removed lithology aspect from the revised version of Table 2.**

L 76 : The authors have not scrutinized the literature sufficiently. There have been direct measurements of CO2 in "Northern Eurasia". Please refer to the work of Castro-Morales et al. (2022) in the Ambolikha River in northeast Siberia, published in December 2021. It is not the job of reviewers to make the literature overview for the authors. This paper should be relevant for the discussion as it also reports diurnal variations.

**We would like to point out that the work of Castro-Morales et al has been cited in both the initial submission and revised version of our manuscript.**

**We made sure that we cited all works relevant to $CO_2$ emissions in Siberian river waters in the newly revised version, and added a few more references. We strongly reorganized the text in the Introduction (L 79-85). Note that we did not analyze in details the vast literature on Scandinavian fluvial systems; this would require a review paper in its own.**

L81 : I still do not understand what you mean by "high latitude regions are important". This statement is so vague it could mean anything.

**We removed this rather vague sentence from the Introduction.**

L83: So, what ? In which form will this carbon be released? Will this carbon release impact atmospheric $CO_2$ or $CH_4$ ? If the organic carbon is released in extremely refractory form and is not converted (or converted super slowly) by microbial activity into $CH_4$ or $CO_2$, then this will have no impact on atmospheric carbon content.

**We removed these lines from the revised text; discussion this issue goes above the objectives of this work. We simply stated that "The on-going interest to Siberia comes from the fact that this region hosts large C stocks in soils and wetlands intersected by extensive river networks that deliver majority of water and C to the Arctic Ocean (Feng et al., 2013)."**

I suggest that the authors make their data-set publically available on publication either as a supplement or an entry in a data-repository (zenodo or equivalent).

**Most of the data are presented in the Supplement (Table S2). However, following this important remark, we uploaded all primary data on continuous $pCO_2$ in the Ket River to the Mendeley database, where we publish all other similar data on Siberian Rivers:**

**Pokrovsky, O., Lim, A., Krickov, I., Korets, M., Vorobyev, S.: "Ket River hydrochemistry, $CO_2$ concentration and landscape parameters", Mendeley Data, V1, doi: 10.17632/snwbkvg6tc.1, 2022.**

**We added necessary 'Data availability' statement in the revised manuscript.**

**We thank the reviewer for very pertinent remarks that allowed significant improvement of the manuscript.**

**Oleg S. Pokrovsky**